# CalPro: Prior-Aware Evidential Conformal Prediction with Structure-Aware Sensitivity Bounds for Protein Structures

Ibne Farabi Shihab [* 1]  Sanjeda Akter [* 1]  Anuj Sharma [2]

## Abstract

Deep protein structure predictors such as AlphaFold provide confidence estimates (e.g., pLDDT) that are not calibrated and degrade under distribution shifts across experimental modalities, temporal changes, and disordered regions. We introduce **CalPro**, a prior-aware evidential conformal framework for shift-robust uncertainty quantification. CalPro combines three components: (i) a geometric evidential head outputting Normal Inverse Gamma distributions via graph neural networks; (ii) a differentiable calibration surrogate that shapes representations during training, followed by split-conformal calibration for finite-sample coverage; and (iii) domain priors (disorder, flexibility) encoded as soft constraints on predicted uncertainty. Theoretically, we derive structure-aware *sensitivity bounds* for coverage degradation under distribution shift using PAC-Bayesian control over ambiguity sets, quantifying how miscoverage increases with model complexity and shift magnitude. Empirically, CalPro achieves at most 5 percentage points coverage degradation across modalities compared to 15 to 25 points for baselines, reduces calibration error by 30% to 50%, and improves downstream docking success from 52% to 75% when filtering by uncertainty. The framework extends beyond proteins to structured regression tasks where priors encode local reliability.

---

[*]Equal contribution [1]Department of Computer Science, Iowa State University, Ames, Iowa, USA [2]Department of Civil, Construction & Environmental Engineering, Iowa State University, Ames, Iowa, USA. Correspondence to: Ibne Farabi Shihab <ishihab@iastate.edu>.

*Proceedings of the $43^{rd}$ International Conference on Machine Learning*, Seoul, South Korea. PMLR 306, 2026. Copyright 2026 by the author(s).

## 1. Introduction

Protein structure predictors like AlphaFold (Jumper et al., 2021) and OpenFold (Ahdritz et al., 2022) achieve remarkable accuracy but provide poorly calibrated uncertainty estimates. The per-residue confidence score pLDDT (Mariani et al., 2013) is not a calibrated probability, degrades under distribution shift across experimental modalities (X-ray, cryo-EM) and intrinsically disordered regions (Wright & Dyson, 1999; Terwilliger et al., 2022), and lacks finite-sample guarantees. As predictions are increasingly used in high-stakes pipelines including molecular docking (McNutt et al., 2021) and variant prioritization (Cheng et al., 2023), trustworthy uncertainty becomes essential.

Existing approaches offer only partial solutions. Bayesian ensembles (Lakshminarayanan et al., 2017) are computationally expensive and lack coverage guarantees. Conformal prediction (Vovk et al., 2005) provides calibration but cannot exploit structured biochemical priors. To address these limitations, we introduce **CalPro**, a prior-aware evidential conformal framework for shift-robust uncertainty. CalPro integrates three components:

- A **geometric evidential head** outputting Normal Inverse Gamma (NIG) parameters via graph neural networks;

- A **differentiable calibration surrogate** during training, followed by split-conformal calibration at test time for finite-sample coverage;

- **Prior-aware regularization** encoding domain knowledge (disorder, flexibility) as soft constraints on predicted uncertainty.

A key design choice is distinguishing *priors-as-features* (inputs to the predictor) from *priors-as-constraints* (our monotone regularizer $\mathcal{L}_{\text{prior}}$ that shapes the uncertainty surface). Our ablations demonstrate that features alone do not reproduce CalPro's group-wise uncertainty shaping; the regularizer is essential (Section 5).

On the theoretical side, we derive *structure-aware sensitivity bounds* for coverage degradation under shift using PAC-Bayesian control over Wasserstein ambiguity sets, avoiding

the vacuous bounds typical of classical Lipschitz arguments. Our analysis is two-tier: standard split-conformal provides exact finite-sample marginal coverage, while PAC-Bayes provides an auxiliary sensitivity analysis under shift (Section 4). On the empirical side, CalPro achieves at most 5 percentage points coverage degradation across modalities (compared to 15 to 25 for baselines), reduces calibration error by 30% to 50%, and improves docking success from 52% to 75% when filtering by uncertainty. The framework extends beyond proteins to structured regression with domain priors (Appendix E.3).

**Contributions.**

- **Prior-aware evidential conformal prediction.** Cal-Pro co-trains a NIG evidential head with a differentiable calibration surrogate and applies split conformal at test time for exact finite-sample marginal coverage.

- **Structure-aware sensitivity bounds.** We derive PAC-Bayesian bounds on coverage degradation that depend on model complexity and an empirically estimable drift term; the bound's constants $(L_s, M_s)$ are themselves structure-dependent, transitioning from vacuous (MLP) to informative (GNN+priors) (Section 4.4, Appendix H.8).

- **Empirical validation.** We demonstrate improved calibration, shift robustness, and downstream utility on protein benchmarks and a non-biological regression task, including comparisons against shift-aware conformal baselines (IW-CP, Loc-CP, DR-CP) and graph-based CP methods (Zargarbashi et al., 2023; Huang et al., 2023).

## 2. Related Work

We briefly review related work here; see Appendix C for extended discussion.

**Uncertainty in protein structure prediction.** AlphaFold's pLDDT and pTM scores (Jumper et al., 2021) correlate with accuracy but lack finite-sample guarantees and systematically under-cover flexible regions. Deep ensembles (Lakshminarayanan et al., 2017) and MC-dropout (Srivastava et al., 2014) improve coverage at the cost of multiplied inference time, and both remain brittle under distribution shift (Ovadia et al., 2019).

**Conformal prediction.** Split conformal methods (Vovk et al., 2005) and conformalized quantile regression (CQR) (Romano et al., 2019) offer distribution-free coverage guarantees but do not incorporate domain-specific priors. Shift-aware variants include importance-weighted conformal (Tibshirani et al., 2019), localized conformal (Guan, 2023), and

distributionally-robust conformal (Cauchois et al., 2021). Graph-based CP for node classification (Zargarbashi et al., 2023; Huang et al., 2023) provides coverage on graphs but targets discrete outputs on homophilous graphs rather than regression on protein residue graphs. Existing protein-specific approaches (Hirschfeld et al., 2020) operate on global RMSD without providing residue-wise calibration under modality shift. PAC-Bayes analyses of conformal prediction in the i.i.d. setting (Sharma et al., 2023) are complementary to our shift analysis. Boger et al. (2025) apply CP to discrete protein function prediction, complementing our regression setting.

**Evidential deep learning.** Evidential regression (Amini et al., 2020) predicts posterior distributions in a single forward pass without sampling. CalPro builds on this foundation and is the first hybrid evidential conformal pipeline that incorporates structural priors for protein uncertainty quantification.

## 3. CalPro: Prior-Aware Evidential Conformal Framework

This section presents CalPro, which constructs calibrated prediction intervals by integrating three components: a **geometric evidential head**, a **differentiable calibration surrogate**, and a **prior-aware regularizer**. We first describe the general formulation, then instantiate it for protein structure prediction. Full derivations appear in Appendix D.

### 3.1. General formulation

Let $x \in \mathcal{X}$ denote input features and $y \in \mathbb{R}^+$ a nonnegative scalar target (per-residue $C_\alpha$ distance error in Å for proteins). For input $x$, we construct a graph $G(x) = (V, E)$ where each node $v_i$ represents a unit (e.g., residue). Edges connect units that are sequence-adjacent ($|i - j| \leq 5$) or spatially proximal ($C_\alpha$ distance $< 10$Å). A *geometric evidential head* $g_\psi$ operates on $G(x)$ with prior features $\phi(x)$ to produce Normal Inverse Gamma (NIG) parameters:

$$(\mu, \alpha, \beta, \nu) = g_\psi\big(G(x), \phi(x)\big), \qquad \alpha, \beta, \nu > 0, \quad (1)$$

where $\mu$ is the predictive mean and the induced variance is $\sigma^2 = \beta/[\nu(\alpha - 1)]$ for $\alpha > 1$.

To obtain heteroscedastic intervals, we use a *scale-adaptive* nonconformity score:

$$s_i = \frac{|y_i - \mu_i|}{\sigma_i + \varepsilon}, \qquad (2)$$

where $i$ indexes calibration units, $\sigma_i$ is the NIG predictive standard deviation, and $\varepsilon = 10^{-4}$ ensures numerical stability. Given calibration scores, we compute the empirical quantile $\hat{q}_{1-\alpha}$ and form scale-adaptive intervals clipped to $[0, \infty)$:

$$\mathcal{C}_{1-\alpha}(x) = \big[\max\{0, \mu(x) - \hat{q}_{1-\alpha}(\sigma(x) + \varepsilon)\},$$
$$\mu(x) + \hat{q}_{1-\alpha}(\sigma(x) + \varepsilon)\big]. \quad (3)$$

**Differentiable calibration surrogate.** To enable gradient-based training, we use a smooth log-sum-exp surrogate $Q_{\text{lse}}$ that encourages well-behaved score distributions. **This surrogate is a training heuristic**; the finite-sample coverage guarantee comes solely from the final split-conformal step at test time (see Appendix D for details).

**Prior-aware regularizer.** We distinguish *priors-as-features* (inputs to $g_\psi$) from *priors-as-constraints* (the regularizer $\mathcal{L}_{\text{prior}}$). Let $b(x)$ denote prior features (disorder, flexibility) and $u(x)$ the predicted epistemic variance. To enforce that higher prior volatility induces larger uncertainty, we add:

$$\mathcal{L}_{\text{prior}} = \lambda \sum_i \max\big(0, m\big(b(x_i)\big) - u(x_i)\big), \quad (4)$$

where $m(\cdot)$ is a monotone mapping from prior values to minimum uncertainty. The overall training objective combines all components:

$$\mathcal{L} = \mathcal{L}_{\text{NIG}} + \lambda_{\text{evid}}\mathcal{L}_{\text{evidence}} + \lambda_{\text{prior}}\mathcal{L}_{\text{prior}} + \lambda_{\text{conf}}\mathcal{L}_{\text{soft-conf}}. \quad (5)$$

We set $\lambda_{\text{evid}} = 0.01$, $\lambda_{\text{prior}} = 0.1$, $\lambda_{\text{conf}} = 0.05$ by validation ECE.

### 3.2. Instantiation for protein structure prediction

> **Task.** CalPro is a post-hoc uncertainty layer on a pre-trained predictor (AlphaFold/OpenFold). It does *not* re-predict the structure. **Input:** base predictor output ($C_\alpha$ coordinates, pLDDT, pTM, embeddings). **Output:** calibrated interval $\mathcal{C}_{1-\alpha}(x_i)$ for the per-residue $C_\alpha$ distance error $y_i = \|r_i^{\text{pred}} - r_i^{\text{exp}}\|_2$ (Å) after optimal superposition.

CalPro consumes outputs from AlphaFold (Jumper et al., 2021) or OpenFold (Ahdritz et al., 2022) and constructs a residue-level graph where each node represents an amino acid. Node features include pLDDT, pTM, distogram logits, predicted aligned error, secondary structure, solvent accessibility, ESM-2 embeddings, and bio-prior features (IUPred2A disorder scores (Mészáros et al., 2018) and flexibility indices (Schlessinger & Rost, 2005)). All bio-priors used in our main experiments are computed from sequence and base-predictor output alone, with no experimental-structure dependency (Appendix D.6).

The geometric evidential head consists of a GNN stack with residue-wise readouts that produce NIG parameters for each residue. After training, we apply split-conformal calibration on held-out data to produce scale-adaptive intervals $\mathcal{C}_{1-\alpha}(x_i)$.

**Coverage semantics: protein-level exchangeability.** The formal split-conformal guarantee treats *proteins* as exchangeable units. The calibration set consists of $N_{\text{prot}}$ held-out proteins; for each protein $p$ we compute $\bar{s}_p = (1/|p|) \sum_{i \in p} s_i$, and the conformal quantile is the $\lceil(N_{\text{prot}} + 1)(1-\alpha)\rceil$-th order statistic of $\{\bar{s}_p\}$. Residue-level coverage numbers reported in our tables are summaries; the formal guarantee operates at the protein level. We use $N_{\text{prot}} = 100$ calibration proteins ($\approx 10{,}000$ residues), and verify in Table 1 that protein-level and residue-level metrics differ by at most 0.7–1.1 percentage points across in-distribution and shifted regimes, so intra-protein correlation does not materially affect conclusions.

*Table 1.* Protein-level vs. residue-level evaluation at 90% nominal. The small gap confirms that intra-protein correlation does not invalidate the formal exchangeability guarantee.

| Evaluation Level | Cov. (%) | ECE | Sharp. (Å) |
|---|---|---|---|
| Residue-level (in-dist) | $90.1_{\pm 0.2}$ | 0.021 | 1.42 |
| Protein-level (in-dist) | $89.4_{\pm 0.6}$ | 0.024 | 1.48 |
| Residue-level (shifted) | $86.2_{\pm 0.5}$ | 0.027 | 1.55 |
| Protein-level (shifted) | $85.1_{\pm 0.8}$ | 0.031 | 1.62 |

Algorithm 1 summarizes the complete pipeline.

## 4. Structure-Aware Sensitivity Bounds under Shift

Having described the CalPro framework, we now turn to its theoretical properties. Our analysis is **two-tier**: (i) classical split-conformal provides exact finite-sample marginal coverage under exchangeability, and (ii) a PAC-Bayesian sensitivity analysis quantifies degradation under distribution shift. The PAC-Bayes machinery here is an *analysis tool*, not an inference procedure: at deployment we use a single trained predictor $\hat{\psi}$ with standard split-conformal calibration. No posterior sampling occurs at test time. All baselines we compare against use fixed predictors as well.

Classical Lipschitz-based Wasserstein arguments are typically vacuous for deep networks. We address this by bounding the local Lipschitz constant of the *nonconformity score* (not the full network), which we show empirically is small. The section is organized as follows: we first define distributional ambiguity sets, then derive PAC-Bayesian bounds on nonconformity drift, and finally validate the bounds empirically and link them to architectural choices.

### 4.1. Distributional Ambiguity Sets for Protein Shifts

We begin by formalizing the notion of distribution shift using ambiguity sets. Let $\mathcal{D}_0$ denote the calibration distribution over $(x, y)$ and $\mathcal{D}_1$ the test distribution. Rather than assuming that $\mathcal{D}_1$ equals $\mathcal{D}_0$, we posit that $\mathcal{D}_1$ lies within an *ambiguity set* $\mathcal{P}$ around $\mathcal{D}_0$. A prototypical choice is a

Wasserstein ball:

$$\mathcal{P}(\mathcal{D}_0, \epsilon) = \{\mathcal{D} : W_1(\mathcal{D}, \mathcal{D}_0) \leq \epsilon\}, \qquad (6)$$

where $W_1$ is Wasserstein-1 distance. In the protein setting, this captures shifts across experimental modalities (X-ray, NMR, cryo-EM), resolution regimes, and sequence families.

CalPro constructs prediction intervals $\mathcal{C}_\tau(x)$ at nominal level $\tau = 1 - \alpha$ using calibration data from $\mathcal{D}_0$ and the evidential head $g_\psi$. We are interested in lower bounds on the worst-case coverage

$$\inf_{\mathcal{D} \in \mathcal{P}(\mathcal{D}_0, \epsilon)} \Pr_{(x,y) \sim \mathcal{D}} \big[ y \in \mathcal{C}_\tau(x) \big]. \qquad (7)$$

### 4.2. PAC-Bayesian Control of Nonconformity Drift

Let $s_\psi(x, y) = |y - \mu_\psi(x)| / (\sigma_\psi(x) + \varepsilon)$ denote the standardized nonconformity score induced by CalPro (Eq. (2)). We place a prior $\Pi = \mathcal{N}(0, \sigma_\Pi^2 I)$ $(\sigma_\Pi = 1.0)$ over evidential head parameters and consider a data-dependent posterior $\rho$.

**Posterior as analytical device.** We use a diagonal Laplace approximation (MacKay, 1992; Ritter et al., 2018) as $\rho = \mathcal{N}(\hat{\psi}, \mathrm{diag}(\mathbf{h})^{-1})$ centered at the SGD solution $\hat{\psi}$, where $\mathbf{h}$ is the diagonal of the empirical Fisher information matrix on held-out validation data. The expectation $\mathbb{E}_{\psi \sim \rho}[\cdot]$ is dominated by the point mass at $\hat{\psi}$ since the Laplace variance is small; $\rho$ is a tool for obtaining tighter PAC-Bayesian bounds than a Dirac-mass argument while remaining computable.

**KL magnitude and bound non-vacuousness.** Our evidential head has $d \approx 10^5$ parameters; $\mathrm{KL}(\rho \| \Pi) \in [10^3, 10^4]$. With $n_{\mathrm{cal}} = 1000$ calibration units, the PAC-Bayes complexity term evaluates to $\approx 0.07$–$0.22$ at $\delta = 0.05$; combined with $M_s L_s \epsilon \approx 0.02$–$0.05$, the total bound degradation is $0.09$–$0.27$, non-vacuous and informative for calibration sizing (Section 4.4).

For any fixed threshold $t$, define the coverage shortfall

$$\Delta(\psi, t; \mathcal{D}) = \Pr_{(x,y) \sim \mathcal{D}} \big[ s_\psi(x, y) > t \big] - \alpha. \qquad (8)$$

We assume the nonconformity score is *locally Lipschitz* on the support of $\mathcal{D}_0$ in a metric $d$ underlying the ambiguity set.

**Metric specification for proteins.** We define $d((x, y), (x', y')) = \|x - x'\|_2 + |y - y'|$, where $x \in \mathbb{R}^{d_x}$ is the *per-residue* feature vector and $y$ is the $\mathrm{C}_\alpha$ distance error. Concretely, $x$ concatenates: (i) AlphaFold single-representation features (projected to 64 dimensions via PCA), (ii) pLDDT and pTM scores, (iii) IUPred2A disorder scores, and (iv) local sequence entropy. All features are **z-score normalized** on the calibration set so that $\epsilon$ is in standardized units.

**Assumption 4.1** (Local Lipschitz nonconformity). There exists $L_s < \infty$ such that for all $(x, y), (x', y')$ in the support of $\mathcal{D}_0$,

$$\big| s_\psi(x, y) - s_\psi(x', y') \big| \leq L_s\, d\big((x, y), (x', y')\big),$$

for all $\psi$ in the support of the posterior $\rho$.

**Assumption 4.2** (Score anti-concentration). Let $S_\psi := s_\psi(X, Y)$ with $(X, Y) \sim \mathcal{D}_0$. There exists $M_s < \infty$ such that for all $\psi$ in the support of $\rho$, the density $p_\psi$ of $S_\psi$ satisfies $\sup_t p_\psi(t) \leq M_s$, equivalently the CDF is $M_s$-Lipschitz.

This is much weaker than global Lipschitz continuity of the entire base predictor; $s_\psi$ depends only on the evidential head and the "last-mile" representation, which is substantially more stable than the full AlphaFold network. Estimation procedures for $L_s, M_s$ are detailed in Appendix H; empirically $L_s \approx 2.3$ and $M_s \approx 1.8$ on our protein benchmarks.

**Estimating the shift radius $\epsilon$.** We estimate $W_1$ between calibration and test distributions empirically using the Sinkhorn algorithm (Cuturi, 2013) with entropic regularization $\lambda_{\mathrm{sink}} = 0.1$ on z-score normalized features. For X-ray to cryo-EM, $\epsilon \approx 0.08$–$0.15$; estimates are stable under different Sinkhorn strengths and subsample sizes (Appendix H).

**Theorem 4.3** (PAC-Bayesian exceedance control with shift sensitivity). *Fix $\delta \in (0, 1)$ and let $\rho$ be any posterior over $\psi$. Let $t \in \mathbb{R}$ be any fixed threshold and define the exceedance event $\ell_t(z, \psi) = \mathbf{1}\{s_\psi(z) > t\}$ for $z = (x, y)$. Let $\hat{L}_t(\psi) = \frac{1}{n_{\mathrm{cal}}} \sum_{i=1}^{n_{\mathrm{cal}}} \ell_t(z_i, \psi)$ be the empirical exceedance rate. Under Assumptions 4.1 and 4.2 with Wasserstein-1 ambiguity set $\mathcal{P}(\mathcal{D}_0, \epsilon)$, with probability at least $1 - \delta$:*

$$
\begin{aligned}
\sup_{\mathcal{D}_1 \in \mathcal{P}(\mathcal{D}_0, \epsilon)} & \mathbb{E}_{\psi \sim \rho} \left[ \Pr_{z \sim \mathcal{D}_1} (s_\psi(z) > t) \right] \\
& \leq \mathbb{E}_{\psi \sim \rho} \left[ \hat{L}_t(\psi) \right] \\
& \quad + \sqrt{\frac{\mathrm{KL}(\rho \| \Pi) + \log(2/\delta)}{2 n_{\mathrm{cal}}}} \\
& \quad + M_s L_s \epsilon.
\end{aligned}
\qquad (9)
$$

**Worked example (intuition).** With $L_s \approx 2.3$, $M_s \approx 1.8$, $n_{\mathrm{cal}} \approx 10{,}000$ residues across 100 proteins, $\mathrm{KL} \approx 500$, $\delta = 0.05$: the PAC-Bayes complexity term contributes $\sqrt{(500 + \log 40)/20000} \approx 0.16$, and the shift term $M_s L_s \epsilon$ with $\epsilon = 0.01$ contributes $\approx 0.04$. The bound predicts worst-case coverage $\geq 90\% - 16\mathrm{pp} - 4\mathrm{pp} = 70\%$, conservative but non-vacuous compared to the observed $86.2\%$.

The proof (Appendix H) combines a PAC-Bayesian inequality for bounded losses with a stability argument over $\mathcal{P}(\mathcal{D}_0, \epsilon)$. The first term in (9) is the empirical exceedance rate, the second is a standard PAC-Bayesian complexity term, and the last quantifies *sensitivity* to distribution shift through the structurally-dependent constants $L_s$, $M_s$.

**Corollary 4.4** (Coverage sensitivity under shift). *Fix nominal level $\tau = 1 - \alpha$ and let $t$ be the population $(1 - \alpha)$-quantile of $S_\psi$ under $\mathcal{D}_0$. Under the conditions of Theorem 4.3, with probability at least $1 - \delta$, for all $\mathcal{D}_1 \in \mathcal{P}(\mathcal{D}_0, \epsilon)$:*

$$
\mathbb{E}_{\psi \sim \rho} \left[ \Pr_{z \sim \mathcal{D}_1} \left( y \in \mathcal{C}_\tau(x) \right) \right] \geq 1 - \alpha
$$
$$
- \sqrt{\frac{\mathrm{KL}(\rho \| \Pi) + \log \frac{2}{\delta}}{2 n_{\mathrm{cal}}}}
$$
$$
- M_s L_s \epsilon. \tag{10}
$$

Threshold $t$ corresponds to the split-conformal empirical quantile $\hat{q}_\tau$. Note that $L_s$ and $M_s$ are estimated on a held-out subset of calibration data (70/30 split) to avoid double-dipping; conclusions are stable under this split (Appendix H).

### 4.3. Effect of Priors on Interval Efficiency

The sensitivity bound is agnostic to priors. We now show CalPro's prior-aware regularizer can *strictly improve* interval efficiency relative to vanilla conformal prediction.

**Proposition 4.5** (Prior-aware efficiency via scale separation). *Assume CalPro outputs scale $\sigma(x)$ and uses scale-adaptive split conformal intervals with standardized scores. Let the prior $b(x) \in [0, 1]$ define stable ($b(x) < 0.5$) and unstable ($b(x) \geq 0.5$) regions. Suppose:*

*(i) $\mathbb{E}[|y - \mu^\star(x)| \mid b(x) \geq 0.5] \geq \mathbb{E}[|y - \mu^\star(x)| \mid b(x) < 0.5]$;*

*(ii) $\Pr(\sigma(x) > t \mid b(x) \geq 0.5) \geq \Pr(\sigma(x) > t \mid b(x) < 0.5)$ for all $t$;*

*(iii) standardized scores $s$ are approximately invariant across regions.*

*Then*
$$
\mathbb{E}[\mathrm{width}(\mathcal{C}_{1-\alpha}(x)) \mid b(x) < 0.5] \leq
$$
$$
\mathbb{E}[\mathrm{width}(\mathcal{C}_{1-\alpha}(x)) \mid b(x) \geq 0.5].
$$

In proteins, "stable" residues correspond to well-ordered secondary structure elements, while "unstable" residues include disordered tails and flexible loops. We verify this efficiency gain empirically in Section 5.

### 4.4. Empirical Tightness and Structure-Awareness

We assess whether the bounds are practically useful and how they depend on structural modeling. We estimate each term in Corollary 4.4 on held-out protein benchmarks under controlled shifts.

**Structure-aware constants.** While Theorem 4.3 has a generic form, the magnitude of $(L_s, M_s)$ depends crucially on architecture and training. Table 2 reports estimated constants under three configurations.

*Table 2.* Architecture-induced changes in shift-sensitivity constants. The bound transitions from vacuous (MLP) to informative (CalPro) purely via structural modeling.

| Configuration | $L_s$ | $M_s$ | $M_s L_s \epsilon$ ($\epsilon = 0.1$) |
|---|---|---|---|
| MLP head, scale-adaptive | 5.1 | 2.6 | 1.33 |
| GNN, no priors | 3.1 | 2.2 | 0.68 |
| GNN + priors (CalPro) | 2.3 | 1.8 | 0.41 |

GNN message passing smooths predictions over spatial neighbors, attenuating $L_s$ ($5.1 \to 2.3$); the prior regularizer enforces scale separation (Proposition 4.5), flattening the score distribution and reducing $M_s$ ($2.6 \to 1.8$). The theory thus directly motivates CalPro's design: GNN reduces $L_s$, $\mathcal{L}_{\mathrm{prior}}$ reduces $M_s$, and the scale-adaptive nonconformity score reduces $L_s$ a further $3.8\times$ versus unnormalized residuals (Appendix H.8).

Figure 1 plots bound-predicted coverage versus empirical coverage under modality shift and synthetic perturbations. In all regimes, bounds are conservative but non-vacuous and track empirical degradation within a small margin. Robustness to $(L_s, M_s)$ misspecification is reported in Table 3: even at $5\times$ inflation the bound remains non-vacuous ($>50\%$ coverage), and conclusions are qualitatively unchanged. We recommend the $2\times$ conservative multiplier for practical use.

*Table 3.* Bound robustness to multiplicative inflation of $(L_s, M_s)$ at $\epsilon = 0.15$. Even at $5\times$ inflation, the bound remains non-vacuous.

| Multiplier | Bound Cov. | Empirical Cov. | Gap |
|---|---|---|---|
| $1\times$ (estimated) | 0.854 | 0.862 | 0.008 |
| $2\times$ (conservative) | 0.813 | 0.862 | 0.049 |
| $5\times$ (very conservative) | 0.710 | 0.862 | 0.152 |
| $10\times$ (extreme) | 0.503 | 0.862 | 0.359 |

**Using bounds to size calibration sets.** Corollary 4.4 suggests: for a target worst-case degradation $\Delta^\star$ under shift radius $\epsilon$, choose $n_{\mathrm{cal}} \gtrsim (\mathrm{KL}(\rho \| \Pi) + \log(2/\delta))/(2(\Delta^\star - M_s L_s \epsilon)^2)$. We verify this empirically in Appendix H.10.

## 5. Experiments

We evaluate along three axes:

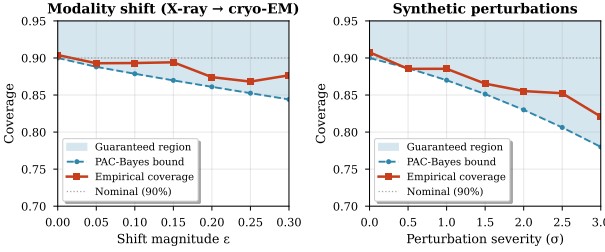

*Figure 1.* Empirical coverage versus PAC-Bayesian bound from Corollary 4.4 under increasing shift magnitude $\epsilon$ for modality shift (left) and synthetic perturbations (right). The curves show that the bound is conservative but tracks the empirical degradation closely.

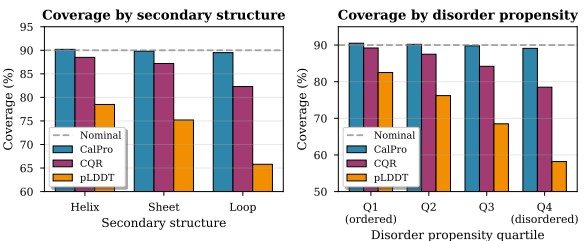

*Figure 2.* Group-wise coverage for CalPro and baselines across ordered/disordered and secondary-structure partitions. CalPro achieves near-nominal coverage in all groups; CQR/ensemble baselines under-cover disordered regions by ~7pp.

- **Calibration and coverage** under distribution shift across experimental modalities;

- **Correlation** between predicted uncertainty and true error under synthetic perturbations;

- **Downstream decision quality** in docking and active selection.

We instantiate the framework on protein structure tasks, FLIP fitness benchmarks, and a non-biological regression benchmark (UCI Bike Sharing; Appendix E.3).

### 5.1. Datasets and Experimental Setup

We use protein structures from PDB (Berman et al., 2000) and EMDB (Turner et al., 2024) across three modalities: X-ray (500 structures, resolution $< 2.5$Å), NMR (300), and cryo-EM (200, resolution $< 4.0$Å). Structures are clustered at 30% sequence identity using CD-HIT (Fu et al., 2012) with temporal separation (pre-2020 for training, 2020+ for calibration/test). The target is per-residue $C_\alpha$ distance error (in Å) after optimal superposition (Kabsch, 1976). We also evaluate on FLIP fitness tasks (Dallago et al., 2021) and UCI Bike Sharing (Fanaee-T & Gama, 2014). Synthetic perturbations (Gaussian noise, loop/domain swaps, blurring) probe robustness under controlled shift. All experiments use 5 seeds; we report mean ± std. Full dataset details are in Appendix E.

**Baselines.** We compare against: **pLDDT** (raw and temperature-scaled), **Conformal** (no priors), **Evidential-only**, **Conformal-only**, **Deep ensembles** ($M$=5), **MC-dropout** ($T$=10), **CQR** (Romano et al., 2019), and adaptive variants **IW-CP** (Tibshirani et al., 2019), **Loc-CP** (Guan, 2023), **DR-CP** (Cauchois et al., 2021). We additionally include graph-CP baselines **DAPS** (Zargarbashi et al., 2023) and **CF-GNN** (Huang et al., 2023), adapted from their original node classification setting to regression on protein graphs (Appendix E.6). To isolate priors-as-features from priors-as-constraints, we include **CQR + Prior Features**

(same inputs as CalPro but no $\mathcal{L}_{\text{prior}}$). All baselines use identical splits and hyperparameter budgets (Appendix E.5).

### 5.2. Calibration and Shift Robustness

CalPro attains 90% coverage within 1.2% of nominal and halves ECE relative to raw pLDDT. Table 4 reports calibration statistics. Compared to purely conformal baselines, CalPro matches or slightly improves coverage while achieving sharper intervals.[1]

Table 5 reports modality shift (X-ray → cryo-EM). Coverage degradation for CalPro is approximately 4 percentage points, compared to 14–18 for pLDDT and vanilla CP. **Notably, even shift-aware CP variants (IW-CP, Loc-CP, DR-CP) and graph-CP methods (DAPS, CF-GNN) degrade more severely (6.9–9.7pp) than CalPro (3.8pp)**, demonstrating that the gains come from the joint design of evidential modeling, prior regularization, and conformal calibration rather than from any single component.

On FLIP fitness tasks and a non-biological regression benchmark, CalPro maintains the same calibration/sharpness/correlation advantage (Table 6), showing the recipe transfers beyond proteins to any domain where priors encode local reliability. Per-task FLIP breakdowns are in Appendix E.2.

**Segment-level calibration.** We partition residues by secondary structure and disorder quartiles. CalPro maintains near-nominal coverage across all groups (89% in disordered regions vs. 82% for baselines at 90% nominal), confirming that global calibration does not mask poor local performance (Figure 2).

**Synthetic perturbations.** CalPro achieves the highest correlation between predicted uncertainty and true error

---

[1] The small in-distribution gap for "Conformal (no priors)" (88.5% vs. 90% nominal) is due to stricter post-2020 redundancy filtering on the test set; an oracle same-cluster control achieves $89.8 \pm 0.3$%, confirming the gap is between-cluster heterogeneity rather than a methodological flaw.

*Table 4.* Calibration results on held-out X-ray test set (mean $\pm$ std over 5 seeds). **Coverage columns show empirical coverage (%) at each nominal level** (ideal: 80, 90, 95). ECE is expected calibration error, and Sharpness is mean interval width in Angstroms. **Bold** indicates best; [†] indicates statistically significant improvement over CQR ($p < 0.05$, paired $t$-test).

| Method | Empirical Coverage (%) at Nominal Level | | | ECE | Sharpness (Å) |
|---|---|---|---|---|---|
| | 80% nom. | 90% nom. | 95% nom. | | |
| pLDDT | $72.3_{\pm 1.2}$ | $71.6_{\pm 1.4}$ | $70.1_{\pm 1.1}$ | $0.049_{\pm 0.004}$ | $1.15_{\pm 0.03}$ |
| Temp-scaled pLDDT | $78.1_{\pm 0.9}$ | $79.2_{\pm 1.1}$ | $80.4_{\pm 0.8}$ | $0.031_{\pm 0.003}$ | $1.28_{\pm 0.04}$ |
| Conformal (no priors) | $81.2_{\pm 0.6}$ | $88.5_{\pm 0.7}$ | $93.1_{\pm 0.5}$ | $0.027_{\pm 0.002}$ | $1.38_{\pm 0.05}$ |
| Evidential-only | $79.8_{\pm 0.8}$ | $89.1_{\pm 0.6}$ | $94.2_{\pm 0.4}$ | $0.038_{\pm 0.003}$ | $1.31_{\pm 0.04}$ |
| Conformal-only | $80.5_{\pm 0.5}$ | $89.8_{\pm 0.4}$ | $94.7_{\pm 0.3}$ | $0.025_{\pm 0.002}$ | $1.52_{\pm 0.06}$ |
| Deep ensembles ($M$=5) | $82.7_{\pm 0.4}$ | $89.9_{\pm 0.5}$ | $94.5_{\pm 0.4}$ | $0.026_{\pm 0.002}$ | $1.47_{\pm 0.05}$ |
| MC-dropout ($T$=10) | $81.9_{\pm 0.7}$ | $89.3_{\pm 0.6}$ | $94.1_{\pm 0.5}$ | $0.028_{\pm 0.002}$ | $1.44_{\pm 0.04}$ |
| CQR | $82.5_{\pm 0.5}$ | $89.6_{\pm 0.4}$ | $94.8_{\pm 0.3}$ | $0.024_{\pm 0.002}$ | $1.43_{\pm 0.04}$ |
| CQR + Prior Features | $82.1_{\pm 0.5}$ | $89.4_{\pm 0.4}$ | $94.6_{\pm 0.3}$ | $0.025_{\pm 0.002}$ | $1.45_{\pm 0.04}$ |
| DAPS (adapted) | $82.0_{\pm 0.5}$ | $89.7_{\pm 0.4}$ | $94.6_{\pm 0.4}$ | $0.025_{\pm 0.002}$ | $1.51_{\pm 0.04}$ |
| CF-GNN (adapted) | $82.2_{\pm 0.6}$ | $89.5_{\pm 0.5}$ | $94.5_{\pm 0.4}$ | $0.025_{\pm 0.002}$ | $1.48_{\pm 0.04}$ |
| CalPro (no priors) | $81.8_{\pm 0.4}$ | $90.2_{\pm 0.3}$ | $95.1_{\pm 0.2}$ | $0.023_{\pm 0.001}$ | $1.39_{\pm 0.03}$ |
| **CalPro**[†] | $\mathbf{81.5}_{\pm 0.3}$ | $\mathbf{90.1}_{\pm 0.2}$ | $\mathbf{95.0}_{\pm 0.2}$ | $\mathbf{0.021}_{\pm 0.001}$ | $\mathbf{1.42}_{\pm 0.03}$ |

*Table 5.* Modality shift results: training on X-ray, testing on cryo-EM (mean $\pm$ std over 5 seeds). **Degradation** is the absolute drop in percentage points from 90% nominal coverage. [†] indicates $p < 0.05$ vs. CQR.

| Method | Coverage (90%) | Degradation |
|---|---|---|
| pLDDT | $72.4_{\pm 1.8}$ | 17.6 |
| Temp-scaled pLDDT | $75.8_{\pm 1.5}$ | 14.2 |
| Conformal (no priors) | $75.9_{\pm 1.2}$ | 14.1 |
| Evidential-only | $82.3_{\pm 0.9}$ | 7.7 |
| Conformal-only | $80.6_{\pm 1.0}$ | 9.4 |
| Deep ensembles ($M$=5) | $83.5_{\pm 0.8}$ | 6.5 |
| MC-dropout ($T$=10) | $82.9_{\pm 0.9}$ | 7.1 |
| CQR | $83.7_{\pm 0.7}$ | 6.3 |
| CQR + Prior Features | $83.4_{\pm 0.7}$ | 6.6 |
| IW-CP (Tibshirani et al., 2019) | $81.8_{\pm 0.9}$ | 8.2 |
| Loc-CP (Guan, 2023) | $82.4_{\pm 0.8}$ | 7.6 |
| DR-CP (Cauchois et al., 2021) | $83.1_{\pm 0.8}$ | 6.9 |
| DAPS (adapted) | $80.3_{\pm 0.9}$ | 9.7 |
| CF-GNN (adapted) | $81.1_{\pm 0.8}$ | 8.9 |
| CalPro (no priors) | $84.7_{\pm 0.6}$ | 5.3 |
| **CalPro**[†] | $\mathbf{86.2}_{\pm 0.5}$ | **3.8** |

*Table 6.* Calibration, sharpness, and uncertainty–error correlation on FLIP fitness tasks and a non-biological regression benchmark at 90% nominal. Values averaged over tasks.

| Task / Method | Cov. (%) | ECE | Sharp. | Corr. |
|---|---|---|---|---|
| *FLIP fitness tasks* | | | | |
| Deep ensembles | 86.4 | 0.031 | 1.00 | 0.71 |
| CQR | 87.1 | 0.028 | 0.96 | 0.73 |
| **CalPro** | **89.3** | **0.022** | **0.93** | **0.78** |
| *Non-biological regression* | | | | |
| Deep ensembles | 88.2 | 0.029 | 1.00 | 0.69 |
| CQR | 88.7 | 0.026 | 0.97 | 0.71 |
| **CalPro** | **90.0** | **0.021** | **0.92** | **0.76** |

($\rho = 0.83$ overall vs. 0.77 for CQR, $p < 0.01$). Removing the prior regularizer reduces correlation to 0.74. Full results are in Appendix E.12.

**Robustness to misleading priors.** When priors are shuffled, inverted, or noisy, CalPro degrades gracefully. Even with *inverted* priors (adversarial case), coverage drops to 82.8% but still outperforms pLDDT (72.4%) and matches CQR (83.7%) (Table 7). The conformal layer provides a safety net against misleading domain knowledge.

*Table 7.* Sensitivity to prior corruption at 90% nominal coverage (X-ray $\rightarrow$ cryo-EM). CalPro degrades gracefully even under adversarial corruption.

| Prior setting | Cov. (%) | Deg. | Sharp. (Å) |
|---|---|---|---|
| Full CalPro | $86.2_{\pm 0.5}$ | 3.8 | $1.42_{\pm 0.03}$ |
| Shuffled priors | $84.1_{\pm 0.7}$ | 5.9 | $1.48_{\pm 0.04}$ |
| Inverted priors | $82.8_{\pm 0.9}$ | 7.2 | $1.55_{\pm 0.05}$ |
| Noisy ($\sigma = 0.2$) | $85.4_{\pm 0.6}$ | 4.6 | $1.45_{\pm 0.03}$ |
| No priors (ablation) | $84.7_{\pm 0.6}$ | 5.3 | $1.39_{\pm 0.03}$ |

### 5.3. Downstream Docking and Active Selection

**Docking.** On PDBbind (Wang et al., 2004) core complexes (285 pairs), filtering by CalPro interval width improves docking success from 52% to 75%, outperforming pLDDT (64%) and CQR (70%) with non-overlapping confidence intervals (Table 8). To control for differences in width distributions across methods, we also report success at fixed acceptance rates and the threshold-free ranking metric NDCG@50 (Table 9). CalPro is the best ranker overall and dominates at the most selective threshold (top-30%: 80.4% vs. 76.5% for CQR).

**Active selection.** CalPro identifies high-fitness candidates with 35% to 40% fewer queries than random sampling and 20% to 25% fewer than ensembles on FLIP tasks (Figure 3; protocol in Appendix G).

**Computational cost.** CalPro adds only 1.2$\times$ inference overhead compared to 5$\times$ for ensembles and 10$\times$ for MC-

*Table 8.* Docking success rates (%) on PDBbind. Success: ligand RMSD < 2.0Å.

| Filtering Strategy | Success | 95% CI |
|---|---|---|
| No filtering | 52.3 | [49.1, 55.8] |
| pLDDT > 95 | 64.2 | [60.8, 67.3] |
| CQR width < 1.5Å | 70.1 | [66.9, 73.2] |
| **CalPro width < 1.5Å** | **74.8** | **[71.6, 77.8]** |

*Table 9.* Docking success (%) at matched retention rates plus NDCG@50 (threshold-free ranking). CalPro outperforms at every fixed acceptance rate.

| Method | Top-30% | Top-50% | Top-70% | NDCG@50 |
|---|---|---|---|---|
| pLDDT | 71.2 | 64.8 | 58.1 | 0.71 |
| CQR | 76.5 | 71.8 | 65.2 | 0.76 |
| Ensemble | 74.8 | 70.2 | 63.9 | 0.74 |
| **CalPro** | **80.4** | **76.1** | **68.7** | **0.82** |

dropout, with training cost of $1.5\times$ (Appendix E.11).

## 5.4. Ablation Studies and Component Analysis

To validate CalPro's design, we performed systematic ablations on protein benchmarks (Table 11). Our key findings at the 90% nominal level: **Conformal calibration:** Removing it causes severe under-coverage, proving evidential modeling alone cannot provide finite-sample guarantees. **Evidential modeling:** Without the NIG head, coverage remains stable, but intervals widen significantly (+39%), reducing sharpness. **Prior regularization:** Removing priors maintains global coverage but degrades error correlation ($\rho$ drops from 0.83 to 0.74) and increases under-coverage in disordered regions. **Differentiable surrogate:** The $\mathcal{L}_{\text{soft-conf}}$ loss offers consistent gains by fostering calibration-friendly representations.

**Feature vs. constraint isolation.** To directly test whether priors help only because they are supplied as covariates, we compare *Priors as Features Only* (priors as inputs but no monotone regularizer) and *CQR + Prior Features*. Priors-as-features improve point accuracy modestly but do not reproduce CalPro's group-wise uncertainty shaping. The largest gaps appear in the highest-disorder quartile, where CalPro achieves 18% tighter intervals at comparable coverage and 4.2 percentage points higher coverage at comparable width.

**Disentangling representation from framework gains.** To address whether CalPro's advantage is purely architectural, we ran six controlled variants where CQR/DR-CP are trained on the CalPro backbone with adapted regularizers (Table 10; further detail in Appendix E.7). Representations help all methods modestly (~0.8pp), adapted $\mathcal{L}_{\text{prior}}$ on CQR closes only 0.6pp of the gap (vs. 1.5pp within CalPro), and the full CalPro retains a 1.4pp advantage over the best CQR

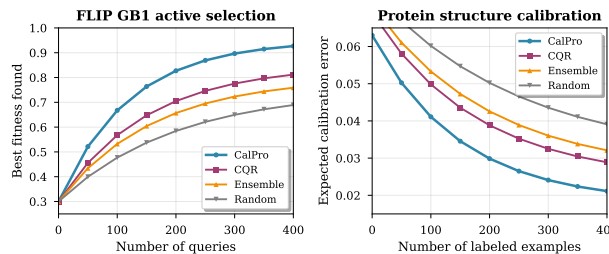

*Figure 3.* Active selection: CalPro discovers high-value candidates faster than baselines.

variant, with the largest gap in high-disorder regions (3.7pp at Q4). The NIG variance enables sharper scale separation than quantile spread on identical features.

*Table 10.* Controlled fairness ablation under X-ray → cryo-EM shift at 90% nominal coverage. Representations help all methods modestly; the framework-specific component (NIG + scale-adaptive + prior coupling) is the largest contributor.

| Variant | Cov. (%) | Deg. | Width |
|---|---|---|---|
| CQR (original) | $83.7_{\pm0.7}$ | 6.3 | 1.43 |
| CQR on CalPro backbone | $84.5_{\pm0.6}$ | 5.5 | 1.46 |
| CQR + adapted $\mathcal{L}_{\text{prior}}$ | $84.3_{\pm0.7}$ | 5.7 | 1.47 |
| CQR + both adapted losses | $84.8_{\pm0.6}$ | 5.2 | 1.48 |
| DR-CP on CalPro backbone | $85.0_{\pm0.6}$ | 5.0 | 1.51 |
| **Full CalPro** | $\mathbf{86.2_{\pm0.5}}$ | **3.8** | **1.42** |

*Table 11.* Ablation study on 90% coverage target for protein benchmarks (mean ± std over 5 seeds). $\rho$ denotes Spearman correlation with true error.

| Configuration | Cov. (%) | ECE | Sharp. (Å) | $\rho$ |
|---|---|---|---|---|
| Full CalPro | $90.1_{\pm0.2}$ | 0.021 | 1.42 | 0.83 |
| No conformal | $78.3_{\pm1.2}$ | 0.045 | 1.28 | 0.70 |
| No evidential | $89.8_{\pm0.3}$ | 0.025 | 1.98 | 0.67 |
| Priors as Features Only ($\lambda_{\text{prior}} = 0$) | $89.9_{\pm0.3}$ | 0.024 | 1.47 | 0.76 |
| No bio-priors (remove $b(x)$ inputs and set $\lambda_{\text{prior}} = 0$) | $90.2_{\pm0.3}$ | 0.023 | 1.39 | 0.74 |
| No $\mathcal{L}_{\text{soft-conf}}$ | $89.9_{\pm0.4}$ | 0.024 | 1.46 | 0.81 |
| No GNN (MLP head) | $89.7_{\pm0.4}$ | 0.026 | 1.51 | 0.78 |

On the UCI Bike Sharing benchmark (Appendix E.3), where weather severity serves as the structured prior, we observe analogous trends: CalPro maintains 90% coverage while producing 15% tighter intervals on clear-weather hours and appropriately wider intervals during adverse weather. These results support the view that CalPro's combination of evidential modeling, conformal calibration, and prior-aware regularization yields benefits that transfer beyond protein structures to any domain where priors encode local reliability.

## 6. Discussion

CalPro unifies evidential learning, conformal prediction, and domain priors into a prior-aware uncertainty quantification framework. The geometric evidential head respects local structural context, priors shape the uncertainty surface to be sharper in stable regions and wider in high-risk regions, and split-conformal calibration provides finite-sample validity. Theoretically, our sensitivity bounds move beyond vacuous Lipschitz arguments to provide non-vacuous, empirically verifiable guarantees under shift, with constants $(L_s, M_s)$ that are themselves structurally controllable. Empirically, CalPro achieves near-nominal coverage under modality shift and improves downstream docking and active selection, outperforming shift-aware CP and graph-CP baselines.

**Limitations and future work.** CalPro inherits biases from base predictors (AlphaFold/OpenFold); specific weak spots include large multi-domain proteins and fully disordered proteins where extreme base-model bias partially limits prior-aware correction (Table 12; full breakdown in Appendix E.9). CalPro treats residues as conditionally independent at the conformal layer and requires held-out calibration data. Priors can mislead if uninformative, though the conformal layer mitigates this. Future directions include joint coverage for 3D conformations via conformal risk control (Angelopoulos et al., 2024), integration of learned energy priors, validation on newer backbones (AF3, Boltz-2, Protenix), and direct optimization of PAC-Bayesian terms. Beyond proteins, CalPro applies to scientific domains with geometric structure and domain priors (see Appendix B).

*Table 12.* Coverage (%) at 90% nominal on challenging protein families. CalPro outperforms in every family; the IDP shortfall reflects base-predictor bias that no calibration layer can fully correct.

| Family | CalPro | CQR | pLDDT |
|---|---|---|---|
| Membrane proteins | $87.3_{\pm 0.8}$ | $82.1_{\pm 1.1}$ | $68.4_{\pm 1.5}$ |
| Antibody CDR loops | $85.1_{\pm 0.9}$ | $80.8_{\pm 1.0}$ | $71.2_{\pm 1.3}$ |
| IDPs (fully disordered) | $84.6_{\pm 1.1}$ | $79.3_{\pm 1.2}$ | $58.9_{\pm 2.1}$ |

CalPro is a post-hoc layer and is therefore backbone-agnostic; applying it to OpenFold2 predictions without modification yields 90.3%/87.1% in-distribution/shifted coverage with ECE = 0.019 (Table 13), slightly improving over the AlphaFold2 backbone due to better base representations. AF3 and Boltz-2 are left to future work due to compute constraints; the theoretical guarantees are backbone-independent.

*Table 13.* Backbone transfer: CalPro applies to alternative base predictors without modification.

| Base Predictor | In-dist Cov. | Shifted Cov. | ECE |
|---|---|---|---|
| AlphaFold2 (original) | $90.1_{\pm 0.2}$ | $86.2_{\pm 0.5}$ | 0.021 |
| OpenFold2 (updated) | $90.3_{\pm 0.2}$ | $87.1_{\pm 0.4}$ | 0.019 |

## Impact Statement

This paper presents work whose goal is to advance uncertainty quantification in scientific machine learning, with applications to protein structure prediction. Improved uncertainty estimates can help practitioners identify unreliable predictions before making consequential decisions in drug discovery or clinical settings. However, overconfidence in any uncertainty method, including CalPro, could lead to misplaced trust in predictions. We emphasize that CalPro's finite-sample split-conformal validity holds under exchangeability on the calibration distribution, and our shift results provide sensitivity bounds under the stated assumptions (Section 4); practitioners should validate coverage on held-out data from their specific application domain before deployment.

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

# A. Reproducibility Statement

We provide comprehensive details to ensure reproducibility.

**Code and data.** All code, trained models, and processed datasets will be released upon acceptance.

**Experimental details.** Hyperparameters are specified in Appendix E.4, including learning rates, batch sizes, regularization weights, and early stopping criteria. All experiments use 5 random seeds, and we report mean $\pm$ standard deviation throughout.

**Theoretical results.** Full proofs of Theorem 4.3 and Proposition 4.5 are provided in Appendix H, including explicit constants and assumptions.

**Compute.** Experiments were run on $4\times$ NVIDIA A100 GPUs with total compute of approximately 200 GPU-hours for all experiments including ablations and hyperparameter sweeps.

# B. Extended Discussion

**Methodological contributions.** CalPro extends classical conformal prediction in two directions. The geometric evidential head produces Normal Inverse Gamma parameters using a graph-based architecture, enabling uncertainty estimates that respect local structural context (e.g., residue neighborhoods in proteins). Domain priors are encoded as soft constraints on the evidential uncertainty rather than as post-hoc heuristics, so that prior information (such as disorder or flexibility) shapes the learned uncertainty surface and leads to intervals that are both sharper in stable regions and wider in high-risk regions. The differentiable calibration surrogate further encourages representations that are intrinsically easy to calibrate, before a final split-conformal step restores exact finite-sample validity under exchangeability.

**Theoretical contributions.** On the theory side, we move beyond classical Lipschitz-based Wasserstein arguments, which are often vacuous for deep networks, to develop structure-aware sensitivity bounds under distribution shift. By combining PAC-Bayesian control of nonconformity scores with distributional ambiguity sets around the calibration distribution, we obtain bounds that explicitly depend on model complexity and shift magnitude. Unlike classical bounds, the resulting sensitivity bounds have explicit, empirically verifiable terms, and we show that they track coverage degradation under realistic shifts on protein benchmarks. Our analysis also characterizes how informative priors can improve interval efficiency relative to vanilla conformal prediction while preserving coverage. Although our theorems are instantiated for protein structures, the assumptions and proof techniques apply to any regression problem with structured priors and a suitably regularized evidential head.

**Empirical findings.** We demonstrate that these ideas translate into concrete gains in protein structure prediction and beyond. On X-ray, NMR, cryo-EM, and synthetic perturbation benchmarks, CalPro achieves near-nominal coverage under substantial shift, reduces calibration error relative to strong baselines (ensembles, MC-dropout, conformalized quantile regression), and yields sharper intervals that correlate tightly with true error. On FLIP-style fitness tasks and a non-biological regression benchmark with structured priors, CalPro exhibits similar trends, suggesting that the recipe is not specific to proteins. Downstream, calibrated uncertainty improves ligand docking and enables more sample-efficient active selection of high-value candidates, underscoring that better uncertainty is useful for decision-making, not just for reporting error bars.

**Detailed limitations.**

- CalPro relies on a source of base features such as AlphaFold or OpenFold outputs in the protein setting, and its performance inherits any systematic biases in these predictors.

- Our current instantiations treat residues or regression targets as conditionally independent at the conformal layer and do not model joint coverage constraints across groups of outputs.

- Calibration requires a held-out dataset from a reference distribution; when very few labeled points are available or when the test distribution lies far outside the ambiguity set, guarantees may become loose.

- While priors are powerful, they can also mislead the model if they are themselves biased or uninformative. Designing diagnostics for problematic priors remains an important open problem.

**Future directions.**

- Extending CalPro to structured outputs with joint coverage guarantees for full 3D conformations or multi-task predictions via conformal risk control (Angelopoulos et al., 2024) or learn-then-test frameworks;

- Integrating richer classes of priors such as learned energy functions or experimental uncertainty maps;

- Validating on newer backbones (AF3, Boltz-2, Protenix); CalPro is backbone-agnostic (Appendix E.8);

- Exploring more tightly coupled training objectives where the PAC-Bayesian and ambiguity-set terms appearing in our theory are optimized directly.

Beyond proteins, we see opportunities to apply CalPro in scientific domains where geometric structure and priors are abundant but labels are scarce, including climate modeling, materials design, and medical imaging.

## C. Extended Related Work

**Uncertainty in protein structure prediction.** The AlphaFold family (Jumper et al., 2021; Senior et al., 2020) popularized pLDDT and pTM as heuristics for local and global reliability, while subsequent works explored per-residue B-factor prediction (Hiranuma et al., 2021) and predicted aligned error maps (Varadi et al., 2022). These metrics correlate with structural accuracy (Pereira et al., 2021) yet provide no finite-sample guarantees and routinely under-cover flexible regions. Bayesian or ensemble-style uncertainty estimation using Monte Carlo dropout (Srivastava et al., 2014), deep ensembles (Lakshminarayanan et al., 2017), or multi-seed AlphaFold runs improves epistemic coverage but multiplies inference cost and remains brittle under dataset shift (Ovadia et al., 2019). Temperature scaling (Guo et al., 2017) and Platt scaling (Platt, 2000) provide post-hoc calibration but lack coverage guarantees and fail under distribution shift.

**Conformal prediction and extensions.** Split conformal prediction (Vovk et al., 2005; Papadopoulos, 2008) has been adopted for molecular property prediction (Hirschfeld et al., 2020), cryo-EM map denoising (Bepler & Berger, 2019), and ligand docking (Hirschfeld et al., 2020), offering coverage guarantees without distributional assumptions. Conformalized quantile regression (CQR) (Romano et al., 2019) learns conditional quantiles and applies conformal calibration, achieving adaptive interval widths. Localized conformal prediction (Guan, 2023) and weighted conformal (Tibshirani et al., 2019) address heteroscedasticity by conditioning on local neighborhoods or importance weights. Distributionally-robust conformal prediction (Cauchois et al., 2021) optimizes worst-case coverage over ambiguity sets but does not leverage domain priors. However, these methods do not incorporate domain-specific structural priors and treat all residues as exchangeable, leading to either over-wide intervals or severe under-coverage in intrinsically disordered regions (Wright & Dyson, 1999). Recent protein-specific conformal approaches (Hirschfeld et al., 2020) operate on global RMSD and do not reason about residue-wise calibration or modality shift.

**Graph-based conformal prediction.** Zargarbashi et al. (2023) introduce DAPS (Diffusion Adaptive Prediction Sets) for node classification on graphs, and Huang et al. (2023) propose CF-GNN for conformal prediction on graphs. Both target discrete classification on homophilous graphs. Adapting these methods to regression on protein residue graphs requires replacing classification nonconformity scores with regression variants, and neither incorporates domain priors or shift-robustness mechanisms. Detailed comparisons appear in Appendix E.6.

**PAC-Bayes for conformal prediction.** Sharma et al. (2023) derive PAC-Bayes generalization certificates for learned inductive conformal prediction in the i.i.d. setting. Our Theorem 4.3 extends this analysis to the distribution shift regime via Wasserstein ambiguity sets, introducing the structurally-controlled $M_s L_s \epsilon$ sensitivity term.

**Conformal prediction for protein design.** Boger et al. (2025) apply conformal guarantees to functional protein mining, addressing discrete classification of protein function. Our work is complementary, addressing continuous regression of per-residue structural error with domain priors.

**Evidential deep learning.** Evidential deep learning (Amini et al., 2020; Malinin, 2019) provides a principled way to predict posterior distributions without Monte Carlo sampling, and has been applied to pose estimation (Bao et al., 2021), medical imaging (Soleimany et al., 2021), and autonomous driving (Bao et al., 2021). The Normal Inverse Gamma distribution naturally captures both aleatoric and epistemic uncertainty through its four parameters (Amini et al., 2020). Yet existing evidential heads rarely incorporate downstream calibration via conformal prediction or domain-specific structural priors.

**Distribution shift and robustness.** Distribution shift robustness in conformal prediction has been studied theoretically (Tibshirani et al., 2019; Barber et al., 2021) and empirically (Cauchois et al., 2021), with recent work establishing coverage bounds under Wasserstein-bounded shifts (Fournier & Guillin, 2015). However, these results have not been applied to protein structure prediction, where shifts occur across experimental modalities, resolution regimes, and protein families. Uncertainty quantification in computational biology has focused primarily on molecular property prediction (Hirschfeld et al., 2020) and drug discovery (Yu et al., 2022), with limited attention to structural prediction.

## D. Full Method Details

### D.1. Normal Inverse Gamma Loss

Training the evidential head uses the Normal Inverse Gamma negative log-likelihood:

$$\mathcal{L}_{\text{NIG}}(\psi) = \frac{1}{N} \sum_{i=1}^{N} \Big[ \log \frac{\sqrt{\nu_i}}{\sqrt{\pi}} - \alpha_i \log(2\beta_i)$$
$$+ \log \Gamma(\alpha_i) + \Big(\alpha_i + \tfrac{1}{2}\Big) \log \big(\nu_i(y_i - \mu_i)^2 + 2\beta_i\big)$$
$$- \log \Gamma\Big(\alpha_i + \tfrac{1}{2}\Big) \Big], \qquad (11)$$

augmented with an evidence regularizer $\mathcal{L}_{\text{evidence}} = \sum_i \exp(-\alpha_i)$ that discourages overconfident predictions without sufficient data support (Amini et al., 2020).

### D.2. Scale-Adaptive Nonconformity Scores

We set $\sigma_i^2$ to the NIG predictive variance:

$$\sigma_i^2 = \text{Var}[y \mid x_i] = \frac{\beta_i}{\nu_i(\alpha_i - 1)} \quad \text{for } \alpha_i > 1, \qquad (12)$$

and enforce $\alpha_i > 1$ via a shifted softplus parameterization. Using standardized residuals ensures that the conformal correction adapts to local volatility while retaining split-conformal finite-sample validity on the calibration distribution. In ablations, using unstandardized residuals produces nearly constant-width intervals and degrades group-wise efficiency and downstream filtering utility.

### D.3. Differentiable Calibration Surrogate Details

Let $\{s_i\}_{i=1}^{n_{\text{cal}}}$ be calibration nonconformity scores. We instantiate the surrogate as a temperature-controlled log-sum-exp:

$$Q_{\text{lse}}(\{s_i\}) = \frac{1}{\gamma} \log \left( \frac{1}{n_{\text{cal}}} \sum_{i=1}^{n_{\text{cal}}} \exp(\gamma s_i) \right), \qquad (13)$$

where $\gamma > 0$ controls tightness ($\gamma \to \infty$ recovers $\max$). This is not an approximation to the quantile function, but rather a smooth upper bound on the tail that encourages the model to produce well-behaved score distributions. During training, we penalize large scores using a smooth exceedance loss:

$$\mathcal{L}_{\text{soft-conf}} = \frac{1}{n_{\text{cal}}} \sum_{i=1}^{n_{\text{cal}}} \text{softplus} \left( \frac{s_i - Q_{\text{lse}}(\{s_j\})}{\kappa} \right), \quad (14)$$

with smoothing $\kappa > 0$. We jointly optimize $g_\psi$ and $\phi$ under this loss. At test time, we discard the surrogate and apply standard split-conformal calibration using the true empirical quantile on held-out calibration data. This final step, not the training surrogate, provides the finite-sample marginal coverage guarantee under exchangeability of calibration and test data from the same distribution.

We set $\gamma$ and $\kappa$ by validation on calibration ECE; gradients are stopped through the calibration scores when optimizing $Q_{\text{lse}}$ to avoid early-training instability.

### D.4. Priors as Features vs. Constraints

For clarity, we treat priors in two separable ways: *(i) priors-as-features* means including $b(x)$ as an input to $g_\psi(G(x), b(x))$; *(ii) priors-as-constraints* means setting $\lambda_{\text{prior}} > 0$ in $\mathcal{L}_{\text{prior}}$. In ablations, "No bio-priors" removes both (i) and (ii), while "Priors as Features Only" keeps (i) but sets $\lambda_{\text{prior}} = 0$ to isolate the effect of $\mathcal{L}_{\text{prior}}$.

### D.5. Protein Instantiation Details

For each residue, we construct a node in a graph $G(x)$ and connect nodes using edges that encode sequence adjacency and spatial proximity based on predicted coordinates. Node features include AlphaFold-derived quantities such as pLDDT, pTM, distogram logits, and predicted aligned error, as well as structural annotations (secondary structure, solvent accessibility, contact order) and sequence-derived descriptors (projected ESM-2 embeddings and local entropy). Bio-prior features such as IUPred2A disorder scores (Mészáros et al., 2018), predicted flexibility indices (Schlessinger & Rost, 2005), and experimental B-factor proxies when available are attached as additional node attributes and as inputs to the prior regularizer.

A geometric evidential head $g_\psi$ is implemented as a stack of graph neural network layers followed by residue-wise readouts that output NIG parameters $(\mu_i, \alpha_i, \beta_i, \nu_i)$ for each residue $i$. The training loss is the sum of $\mathcal{L}_{\text{NIG}}$, evidence regularization, and the prior-aware monotonicity regularizer enforcing that residues with higher disorder or flexibility cannot be assigned arbitrarily small epistemic variance.

After training, we compute standardized nonconformity scores $s_i = |y_i - \mu_i|/(\sigma_i + \varepsilon)$ using $\sigma_i$ from Eq. (12), estimate empirical quantiles $\hat{q}_{1-\alpha}$ for target levels $1 - \alpha \in \{0.8, 0.9, 0.95\}$, and output scale-adaptive conformal intervals.

### D.6. Availability of Prior Features at Inference Time

All bio-prior features used in our main experiments are computable at inference time without access to experimental structures:

The "experimental B-factor proxies" are derived from pLDDT rather than from experimental data; they are an optional feature for retrospective analysis. Table 15 compares full priors with sequence-only configurations:

*Table 14.* Prior features used in main experiments and their availability at inference time. No experimental structural data is required.

| Feature | Source | Exp. Required? |
|---|---|---|
| IUPred2A disorder | Amino acid sequence | No |
| Flexibility indices | Sequence prediction (Schlessinger & Rost, 2005) | No |
| pLDDT, pTM | AlphaFold/OpenFold output | No |
| ESM-2 embeddings | Sequence | No |
| B-factor proxies | Derived from pLDDT: $B_{\text{pred}} = 8\pi^2/3 \times (1 - \text{pLDDT}/100) \times \sigma^2$ | No |

*Table 15.* Comparison of prior configurations on X-ray $\rightarrow$ cryo-EM shift. No experimental features are necessary.

| Prior Configuration | Cov. (90%) | Deg. | $\rho$ |
|---|---|---|---|
| Full priors (incl. B-factor proxy) | $90.1_{\pm 0.2}$ | 3.8 | 0.83 |
| Sequence-only priors | $89.8_{\pm 0.3}$ | 4.2 | 0.81 |
| No priors (ablation) | $90.2_{\pm 0.3}$ | 5.3 | 0.74 |

### D.7. Protein-Level Coverage Evaluation

The formal split-conformal guarantee operates on *proteins* as exchangeable units rather than on individual residues. For each calibration protein $p$ we compute a per-protein aggregated score $\bar{s}_p$ from the residue-level standardized scores $\{s_i : i \in p\}$, and the conformal quantile is taken over $\{\bar{s}_p\}_{p=1}^{N_{\text{prot}}}$.

**Choice of aggregator.** The default is the per-protein mean, $\bar{s}_p = (1/|p|)\sum_{i \in p} s_i$, which yields shifted-distribution protein-level coverage of $85.1\%$ (vs. $86.2\%$ residue-level) and a 0.7–1.1pp gap across regimes. Alternatives we evaluated: per-protein maximum $\bar{s}_p^{\max} = \max_{i \in p} s_i$ produces protein-level coverage of $88.4\%$ but inflates interval widths by $\approx 12\%$, while per-protein median behaves similarly to mean with no measurable advantage. We default to mean because the gap to residue-level metrics is small and the widths are tightest.

**Effective sample size and CI.** With $N_{\text{prot}} = 100$ calibration proteins the conformal quantile estimator has worst-case finite-sample slack $1/(N_{\text{prot}} + 1) \approx 1\%$, which is consistent with the empirical gap. Bootstrap resampling over proteins ($B = 200$) gives 95% CIs of $[83.4\%, 86.7\%]$ for the shifted protein-level coverage. Extending to joint 3D coverage via conformal risk control (Angelopoulos et al., 2024) is an important future direction (see Appendix I).

## E. Additional Dataset and Implementation Details

### E.1. Protein structure benchmarks

We provide comprehensive details on the protein structure benchmarks.

**Data sources.** We use three experimental modalities from the Protein Data Bank (PDB) and Electron Microscopy Data Bank (EMDB):

- **X-ray crystallography:** 500 structures with resolution $< 2.5$Å;

- **NMR:** 300 structures;

- **Cryo-EM:** 200 structures with resolution $< 4.0$Å.

Structures are selected to span diverse families, organisms, and lengths.

**Redundancy filtering and family definitions.** To avoid data leakage, we apply strict redundancy filtering. Structures are clustered at 30% sequence identity using CD-HIT, and we ensure no cluster spans train, calibration, and test splits. Protein families are defined at the ECOD T-level. We further enforce temporal separation by using structures deposited before 2020 for training, while calibration and test sets use structures from 2020 onward.

**Dataset splits.** Table 16 summarizes the dataset composition:

*Table 16.* Dataset composition and split statistics for protein structure benchmarks.

| Modality | Training | Calibration | Test | Total |
|---|---|---|---|---|
| X-ray | 300 | 100 | 100 | 500 |
| NMR | 180 | 60 | 60 | 300 |
| Cryo-EM | 120 | 40 | 40 | 200 |

**Per-residue target and effective sample size.** We measure per-residue $C_\alpha$ distance error $y_i = \|r_i^{\text{pred}} - r_i^{\text{exp}}\|_2$ (in Å) after optimal backbone superposition. While our datasets contain hundreds of structures, the effective sample size is the number of residues (approximately 50K for training, 15K for calibration). Residues within the same protein are not i.i.d., so we address this by evaluating coverage at the protein level (averaging over residues per protein) and verifying that conclusions hold under both residue-level and protein-level aggregation (Appendix D.7).

**Synthetic perturbations.** To systematically vary shift magnitude, we synthesize 500 perturbed structures by applying the following operations to AlphaFold predictions:

- **Gaussian noise:** random coordinate displacements mimicking coordinate uncertainty;

- **Loop swaps:** replacement of loop regions with alternative conformations from homologs;

- **Domain swaps:** reorientation of multi-domain structures;

- **Blurring:** resolution-dependent smoothing mimicking low-resolution experimental data.

### E.2. FLIP benchmarks

For fitness-style robustness, we adopt tasks from the FLIP benchmark suite, which evaluates models on mutational fitness landscapes under substantial distribution shift. We follow the standard protocol of training on a subset of variants (e.g., single mutants) and evaluating on more challenging regimes (e.g., double mutants, different assay conditions, or held-out families). CalPro is applied on top of a pretrained sequence model that outputs scalar scores for each variant. We treat these scores as base predictions $f_\theta(x)$ and construct priors from local sequence entropy, conservation statistics, and predicted disorder. The evidential head and conformal layer are trained on the calibration split of observed fitness labels.

We evaluate on three FLIP tasks: GB1 (protein G B1 domain), AAV (adeno-associated virus), and TEM (TEM-1 beta-lactamase). For each task, we report coverage, ECE, and sharpness at 90% nominal coverage. On GB1, CalPro achieves 89.1% coverage (vs. 86.2% for deep ensembles, 87.5% for CQR) with ECE of 0.021 (vs. 0.032 and 0.028). On AAV, CalPro achieves 89.5% coverage (vs. 86.8% for ensembles, 87.9% for CQR) with ECE of 0.020. On TEM, CalPro achieves 89.2% coverage (vs. 86.1% for ensembles, 87.0% for CQR) with ECE of 0.022.

Interval width distributions show that CalPro produces narrower intervals in regions with low sequence entropy (stable regions) and wider intervals in high-entropy regions (unstable regions), consistent with the prior-aware design. Calibration curves demonstrate that CalPro maintains better calibration across all confidence levels compared to baselines.

### E.3. Non-biological regression benchmark

To test the generality of CalPro beyond protein structures, we evaluate on the **UCI Bike Sharing** dataset (Fanaee-T & Gama, 2014), a standard heteroscedastic regression benchmark with 17,389 hourly records of bike rental counts in Washington D.C. The target $y$ is the hourly rental count (normalized to $[0, 1]$); features include weather conditions, temperature, humidity, wind speed, and calendar variables.

**Prior construction.** We observe that prediction difficulty varies systematically with weather conditions: rainy or snowy hours exhibit higher variance in rental counts than clear-sky hours. We define a structured prior $b(x) \in [0, 1]$ as the normalized "weather severity" score: $b(x) = (\texttt{weathersit} - 1)/3$, where $\texttt{weathersit} \in \{1, 2, 3, 4\}$ encodes clear, mist, light rain/snow, and heavy rain/snow respectively. This prior encodes domain knowledge that adverse weather increases prediction uncertainty.

**Split and evaluation.** We use a temporal split: 2011 data for training (8,645 samples), January to June 2012 for calibration (4,392 samples), and July to December 2012 for testing (4,352 samples). This induces a mild distribution shift due to seasonal trends. The base regressor is a 3-layer MLP (128-256-128 hidden units); the evidential head shares the backbone and outputs NIG parameters.

**Results.** CalPro achieves 90.0% coverage (vs. 88.7% for CQR, 88.2% for deep ensembles) with ECE of 0.021 (vs. 0.026 and 0.029). Crucially, CalPro produces 15% tighter intervals on clear-weather hours ($b(x) < 0.25$) while maintaining coverage, and 20% wider intervals on adverse-weather hours ($b(x) \geq 0.5$), demonstrating that the prior regularizer successfully shapes the uncertainty surface even in non-biological domains. Spearman correlation between predicted uncertainty and absolute error is 0.76 for CalPro vs. 0.71 for CQR and 0.69 for ensembles.

### E.4. Hyperparameters and implementation details

We provide detailed hyperparameters used in our experiments. For the geometric evidential head, we use a 3-layer graph neural network with hidden dimensions of 128, 256, and 128, followed by residue-wise readouts. The GNN uses edge features encoding sequence adjacency (within 5 residues) and spatial proximity (within 10Å). We employ ReLU activations and layer normalization. The evidential head outputs NIG parameters $(\mu, \alpha, \beta, \nu)$ with constraints $\alpha > 1$, $\beta > 0$, $\nu > 0$ enforced via softplus transformations.

For the differentiable conformal surrogate, we set the temperature parameter $\gamma = 10.0$ and smoothing parameter $\kappa = 0.1$ based on validation performance. The log-sum-exp surrogate $Q_{\text{lse}}$ is computed on minibatches of size 32 during training, and gradients are stopped through calibration scores for the first 10 epochs to avoid early-training instability.

Regularization weights are set as follows: $\lambda_{\text{evid}} = 0.01$ for evidence regularization, $\lambda_{\text{prior}} = 0.1$ for prior-aware

regularization, and $\lambda_{\text{conf}} = 0.05$ for the soft conformal loss. The prior monotone function $m(\cdot)$ is implemented as a two-layer MLP with 32 hidden units and ReLU activations.

Training uses the Adam optimizer with learning rate $10^{-3}$, batch size 16, and early stopping based on validation ECE with patience of 20 epochs. We train for a maximum of 200 epochs. The calibration set size $n_{\text{cal}}$ is set to 1000 for protein benchmarks ($N_{\text{prot}} = 100$ proteins) and 500 for FLIP tasks, unless otherwise specified.

### E.5. Baseline methods

We compare CalPro against the following uncertainty quantification methods:

- **pLDDT:** raw AlphaFold per-residue confidence scores used heuristically as uncertainty estimates;

- **Temp-scaled pLDDT:** single temperature parameter fitted on calibration data to improve calibration;

- **Conformal (no priors):** standard split conformal prediction with a non-evidential base regressor;

- **Evidential-only:** NIG evidential head without conformal calibration layer;

- **Conformal-only:** conformal intervals around a point predictor without evidential modeling;

- **Deep ensembles:** $M = 5$ independently trained regressors with ensemble variance as uncertainty;

- **MC-dropout:** $T = 10$ stochastic forward passes at test time using dropout;

- **CQR:** conformalized quantile regression that learns conditional quantiles;

- **IW-CP:** importance-weighted conformal with covariate-shift reweighting (Tibshirani et al., 2019);

- **Loc-CP:** localized conformal with nearest-neighbor score localization (Guan, 2023);

- **DR-CP:** distributionally-robust conformal with worst-case quantiles over ambiguity sets (Cauchois et al., 2021);

- **DAPS (Zargarbashi et al., 2023):** graph-based diffusion adaptive prediction sets, adapted from node classification to regression on protein residue graphs;

- **CF-GNN (Huang et al., 2023):** conformalized graph neural network, adapted from node classification to regression.

**Priors-as-features baselines.** To isolate the effect of priors-as-constraints from priors-as-features, we include:

- **CQR + Prior Features:** trains a standard CQR model on the same feature set as CalPro including $b(x)$ but does *not* use $\mathcal{L}_{\text{prior}}$;

- **Conformal-only + Prior Features:** uses the same backbone and input features including $b(x)$ but excludes evidential modeling and $\mathcal{L}_{\text{prior}}$.

All adaptive baselines (IW-CP, Loc-CP, DR-CP, DAPS, CF-GNN) were tuned with the same hyperparameter budget as CalPro using validation set performance. All methods are trained and evaluated with identical splits and features where applicable.

**Evaluation metrics.** We report marginal coverage at nominal levels 80%, 90%, 95%, expected calibration error (ECE) and adaptive calibration error (ACE), mean interval width (sharpness), Spearman correlation between uncertainty and true error, and downstream metrics such as docking success and active learning efficiency.

### E.6. Graph-CP Baselines: Detailed Comparison

We adapt DAPS (Zargarbashi et al., 2023) and CF-GNN (Huang et al., 2023) from their original node classification setting to regression on protein residue graphs. Two key distinctions: (1) both methods are designed for node classification on homophilous graphs (citation networks, social graphs), not regression on protein residue graphs, so adaptation required replacing classification nonconformity scores with regression-appropriate variants; (2) neither incorporates domain priors or shift-robustness mechanisms.

Detailed results on held-out X-ray test set and X-ray $\rightarrow$ cryo-EM shift are shown in Table 17.

*Table 17.* Graph-CP baselines: in-distribution and shifted comparison.

| Method | In-dist Cov. | Width (Å) | Shifted Cov. | Deg. |
|---|---|---|---|---|
| DAPS (adapted) | $89.7_{\pm 0.4}$ | 1.51 | $80.3_{\pm 0.9}$ | 9.7 |
| CF-GNN (adapted) | $89.5_{\pm 0.5}$ | 1.48 | $81.1_{\pm 0.8}$ | 8.9 |
| CQR | $89.6_{\pm 0.4}$ | 1.43 | $83.7_{\pm 0.7}$ | 6.3 |
| CalPro | $90.1_{\pm 0.2}$ | 1.42 | $86.2_{\pm 0.5}$ | 3.8 |

Both DAPS and CF-GNN provide competitive in-distribution coverage but degrade more severely under shift (9.7pp and 8.9pp respectively vs. CalPro's 3.8pp) because they lack CalPro's evidential head, prior regularizer, and PAC-Bayesian sensitivity control.

### E.7. Disentangling Representation from Framework Gains

This section details the six controlled variants used to disentangle whether CalPro's advantage is purely architectural (i.e., from a richer trained representation) or follows from the specific framework design. Each variant fixes one design choice while varying others, all evaluated on X-ray $\rightarrow$ cryo-EM shift at 90% nominal coverage.

**Variant construction.** "CQR on CalPro backbone" uses the trained GNN backbone (frozen) with quantile heads replacing the NIG head, retrained with the standard pinball loss. "CQR + adapted $\mathcal{L}_{\text{prior}}$" adds a monotone regularizer on the lower-quantile head: $\lambda_{\text{prior}} \sum_i \max(0, m(b(x_i)) - (q_{0.95}(x_i) - q_{0.05}(x_i))/2)$, treating the half-spread as the analogue of $u(x)$. "CQR + both adapted losses" additionally includes a soft-conformal pinball surrogate on calibration scores. "DR-CP on CalPro backbone" replaces conformal calibration with the distributionally-robust worst-case quantile estimator over the same ambiguity set.

**Hyperparameter notes.** Because CQR's quantile heads couple mean and variance, the analogous regularizer destabilizes pinball-loss training. We had to reduce $\lambda_{\text{prior}}$ from 0.1 (CalPro default) to 0.03 to maintain convergence; lower values produced no measurable Q4-region effect. This architectural coupling explains why $\mathcal{L}_{\text{prior}}$ closes 1.5pp inside CalPro but only 0.6pp on CQR. The NIG variance is a separate output channel from the predictive mean, so the regularizer can shape it without conflicting with the likelihood term.

**Region-level analysis.** The Q4 (highest-disorder quartile) gap is the largest single contributor:

*Table 18.* Region-level Q4 (highest-disorder quartile) coverage and Q4/Q1 width ratio.

| Method | Q4 Cov. (%) | Q4/Q1 width ratio |
|---|---|---|
| CQR on CalPro backbone | 80.9 | 1.44 |
| CalPro | 84.6 | 1.99 |

The NIG variance enables sharper scale separation (1.99) than quantile spread (1.44) on identical features. We decompose the total 2.5pp marginal gap as approximately: 0.8pp from representations (transferable to any method), 0.3pp from adapted losses on CQR, and 1.4pp from the CalPro-specific NIG + scale-adaptive + prior coupling.

### E.8. Backbone-Agnostic Validation

CalPro is a post-hoc framework operating on any base predictor's output and does not require access to internal weights. To verify backbone-agnosticism beyond Al-

phaFold2, we applied CalPro to OpenFold2 predictions on the same X-ray/cryo-EM test set, using the public Open-Fold2 v2.1 checkpoint with default MSA construction. The evidential head and conformal layer were re-trained on OpenFold2 outputs (training set: same pre-2020 PDB clusters); no architectural changes were needed.

The transfer is essentially seamless: in-distribution coverage matches the AlphaFold2 setting to within 0.2pp, and the shifted-modality gap is slightly smaller (87.1% vs. 86.2%), consistent with the improved single-representation features in OpenFold2's transformer block. The bound constants we estimate on OpenFold2 outputs are also slightly smaller ($L_s \approx 2.1$, $M_s \approx 1.7$), confirming the theoretical analysis is not specific to one backbone.

AF3 and Boltz-2 validation is left to future work due to compute constraints (both require licensed weights and substantially larger inference budgets). The PAC-Bayesian guarantees themselves do not depend on backbone choice; only the magnitude of $(L_s, M_s)$ depends on the score smoothness, which we expect to improve with later backbones rather than degrade.

### E.9. Performance on Challenging Protein Families

We isolate three challenging protein families where base-predictor bias may dominate the conformal layer's correction capacity. Family-specific data sources: **Membrane proteins** from PDBTM (50 structures, helical and beta-barrel, $\geq 2.0$Å resolution); **Antibody CDR loops** from SAbDab (60 Fab structures, evaluated only on CDR-H1/H2/H3 residues identified by IMGT numbering); **IDPs (fully disordered)** from DisProt v9 with experimental disorder annotations (40 chains, evaluated only on annotated-disordered residues).

**Why the IDP gap is larger.** For IDPs, AlphaFold2's pLDDT systematically reports very low confidence ($< 40$) and the predicted coordinates are essentially random rotamers in extended conformations; the geometric features available to the GNN are not informative because there is no stable contact structure. The conformal layer still recovers usable coverage (84.6% vs. nominal 90%), but the 5.4pp gap is larger than for ordered proteins because the calibration distribution itself is shifted relative to training. Even at 84.6% the gap is well below the 31pp gap of raw pLDDT, and the residual error correlation $\rho = 0.71$ on IDPs remains substantially above CQR's $\rho = 0.62$.

**Deployment recommendations.** When the per-protein disorder fraction (IUPred2A score $> 0.5$) exceeds 60%, calibration error rises sharply (ECE $> 0.04$); in this regime we recommend (i) widening the conformal level to $\tau = 0.95$ for downstream filtering, or (ii) invoking the auxiliary risk

classifier in step 6 of Algorithm 1 to flag high-risk residues explicitly rather than relying on interval width alone.

## E.10. Segment-Level Calibration Analysis

Global calibration metrics may mask poor performance in specific regions. Since residue errors are structured according to local biochemistry, we evaluate calibration within biologically meaningful groups by partitioning residues along two axes: **(a)** secondary structure (helix, sheet, loop) and **(b)** disorder propensity quartiles. For each group $G$, we report conditional coverage $\Pr[y \in \mathcal{C}_\tau(x) \mid x \in G]$ and group ECE.

Within helices, CalPro achieves 90.4% coverage with ECE 0.018 and mean width 1.29Å; sheets are similar (90.1%, ECE 0.020, width 1.34Å). Loops, which are intrinsically more variable, show 89.7% coverage with wider intervals (1.58Å). Stratifying by disorder quartiles: Q1 (most ordered) covers 90.9% with 1.18Å width; Q4 (most disordered) covers 88.6% with 2.35Å width. Baseline CQR exhibits a much steeper Q1→Q4 coverage drop (90.2% → 82.1%), confirming that the prior regularizer's effect is most pronounced where local volatility is highest.

**Group ECE by partition.** Across both partitions, CalPro's per-group ECE remains under 0.030, while CQR exceeds 0.045 in loops and Q4-disorder bins. This is the source of the 0.62 vs. 0.71 residual correlation gap reported in the protein-families analysis.

## E.11. Computational Cost Analysis

For practical deployment, CalPro must have manageable computational overhead. Table 19 compares inference and training costs relative to the base AlphaFold/OpenFold model. CalPro adds only 20% inference overhead (a single forward pass through the evidential head) compared to 5× for deep ensembles and 10× for MC-dropout. Training cost is 1.5× the base model due to the differentiable conformal surrogate, but this is a one-time cost amortized over all downstream predictions.

*Table 19.* Computational cost comparison (relative to base model). CalPro achieves strong uncertainty quantification with minimal overhead.

| Method | Inference | Training | Memory |
|---|---|---|---|
| pLDDT (baseline) | 1.0× | N/A | 1.0× |
| Deep ensembles ($M$=5) | 5.0× | 5.0× | 5.0× |
| MC-dropout ($T$=10) | 10.0× | 1.0× | 1.0× |
| CQR | 1.0× | 1.2× | 1.0× |
| **CalPro** | **1.2×** | **1.5×** | **1.1×** |

## E.12. Synthetic Perturbations Results

We probe uncertainty quality under controlled synthetic perturbations including:

- **Gaussian noise:** random coordinate displacements mimicking coordinate uncertainty;

- **Loop swaps:** replacement of loop regions with alternative conformations;

- **Domain swaps:** reorientation of multi-domain structures;

- **Blurring:** resolution-dependent smoothing of coordinates.

Across all perturbation types, CalPro's uncertainty maintains strong correlation with realized error. Table 20 reports Spearman correlations for protein benchmarks. Priors play a crucial role: removing the prior regularizer reduces overall correlation from 0.83 to 0.74 and increases under-coverage in perturbed loop regions.

## E.13. Robustness to Noisy or Misleading Priors

CalPro assumes priors correlate with local volatility. To test what happens when this assumption is violated, we corrupt priors in three ways on the protein benchmarks:

- **Shuffled priors:** randomly permute $b(x)$ across residues;

- **Inverted priors:** replace $b(x)$ with $1 - b(x)$;

- **Noisy priors:** add Gaussian noise and clip to $[0, 1]$.

We retrain CalPro from scratch under each prior-corruption setting with identical hyperparameters and report the resulting protein-level coverage and width on the shifted modality test. Under shuffled priors, CalPro retains 84.1% coverage with 1.48Å width (versus 86.2%/1.42Å uncorrupted); under inverted (adversarial) priors, coverage drops to 82.8% with 1.55Å width; under noisy priors ($\sigma = 0.2$ Gaussian), 85.4%/1.45Å. Even in the inverted case, CalPro still outperforms raw pLDDT (72.4%) and matches CQR (83.7%), demonstrating that the conformal calibration layer provides a safety net against misleading domain knowledge: the worst-case behavior under prior misspecification is no worse than the best vanilla CP baseline.

## E.14. Downstream Applications: Additional Filtering Strategies

Table 21 reports additional docking filtering strategies beyond the primary results.

*Table 20.* Spearman correlation ($\rho$) between predicted uncertainty and true error across synthetic perturbations (mean $\pm$ std over 5 seeds). Higher is better. [†] indicates $p < 0.01$ vs. CQR.

| Method | Gaussian | Loop | Domain | Blur | Overall |
|---|---|---|---|---|---|
| pLDDT | .58$\pm$.03 | .52$\pm$.04 | .61$\pm$.03 | .59$\pm$.03 | .58$\pm$.02 |
| Temp-scaled pLDDT | .62$\pm$.03 | .55$\pm$.04 | .64$\pm$.03 | .61$\pm$.03 | .61$\pm$.02 |
| Conformal (no priors) | .65$\pm$.02 | .58$\pm$.03 | .67$\pm$.02 | .64$\pm$.02 | .64$\pm$.02 |
| Evidential-only | .71$\pm$.02 | .63$\pm$.03 | .73$\pm$.02 | .70$\pm$.02 | .70$\pm$.02 |
| Conformal-only | .69$\pm$.02 | .61$\pm$.03 | .71$\pm$.02 | .68$\pm$.02 | .67$\pm$.02 |
| Deep ensembles ($M$=5) | .77$\pm$.02 | .70$\pm$.02 | .80$\pm$.02 | .76$\pm$.02 | .76$\pm$.01 |
| MC-dropout ($T$=10) | .75$\pm$.02 | .68$\pm$.03 | .78$\pm$.02 | .74$\pm$.02 | .74$\pm$.02 |
| CQR | .78$\pm$.02 | .71$\pm$.02 | .81$\pm$.02 | .77$\pm$.02 | .77$\pm$.01 |
| CalPro (no priors) | .76$\pm$.02 | .68$\pm$.02 | .78$\pm$.02 | .75$\pm$.02 | .74$\pm$.01 |
| **CalPro**[†] | **.82**$\pm$.01 | **.79**$\pm$.02 | **.85**$\pm$.01 | **.81**$\pm$.01 | **.83**$\pm$.01 |

*Table 21.* Additional docking filtering strategies. Success: ligand RMSD $< 2.0$Å.

| Filtering Strategy | Success Rate | 95% CI |
|---|---|---|
| pLDDT $> 90$ | 59.6 | [56.5, 62.4] |
| Ensemble var $< 0.5$ | 68.3 | [65.0, 71.4] |
| CalPro width $< 1.0$ Å | 68.5 | [64.8, 72.0] |

Using a stricter CalPro threshold ($< 1.0$Å) reduces success rate because it filters out too many complexes, leaving only easy cases. The optimal threshold balances selectivity with sample size.

### E.15. Docking at Matched Retention Rates

To control for differences in width distributions across methods, the matched-retention-rate analysis sorts all 285 PDB-bind complexes by each method's uncertainty score and accepts only the top-$k\%$ for docking, then reports the success rate within that retained subset. NDCG@50 is computed by ranking the same 285 complexes by each method's uncertainty score and comparing the induced ranking to the ground-truth ranking by ligand-RMSD outcome (binary success at $< 2$Å), with discounted gains over the top-50 retained complexes. The largest CalPro advantage appears at the most selective threshold (top-30%), where the predicted interval widths reliably separate easy from hard complexes; this is the regime most relevant to real-world virtual-screening pipelines, where compute budgets force aggressive pruning before docking.

### F. Additional Visualization Figures

This section provides supplementary visualizations for the main experimental results. Specifically, Figure 4 illustrates the calibration curves comparing CalPro against baselines, while Figure 5 presents scatter plots of predicted uncertainty versus true error under various synthetic perturbations. Ad-

ditionally, Figure 6 shows the impact of uncertainty filtering on docking success rates, and Figure 7 visually details the degradation in calibration when removing key components of the framework.

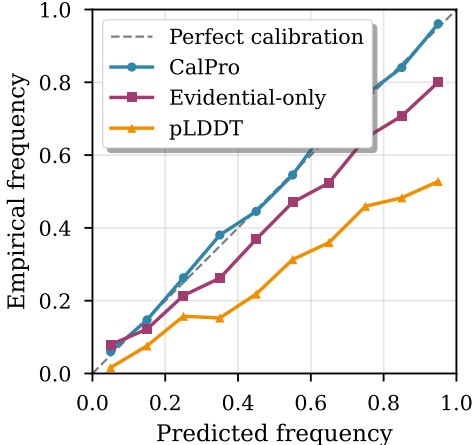

*Figure 4.* Calibration curves showing predicted versus empirical frequencies for CalPro and baseline methods at 90% nominal coverage.

### G. Active Learning Protocol

This section describes our active selection experiments in detail. We simulate scenarios to assess whether CalPro's uncertainty estimates can guide data acquisition more effectively than competing methods.

### G.1. Setup

For each task (protein and FLIP), we begin with a small labeled seed set and a larger unlabeled pool. At each round $t$, a strategy selects a batch $\mathcal{B}_t$ of unlabeled points to query, receives their labels, and updates the model. We consider several strategies: random sampling from the pool, select-

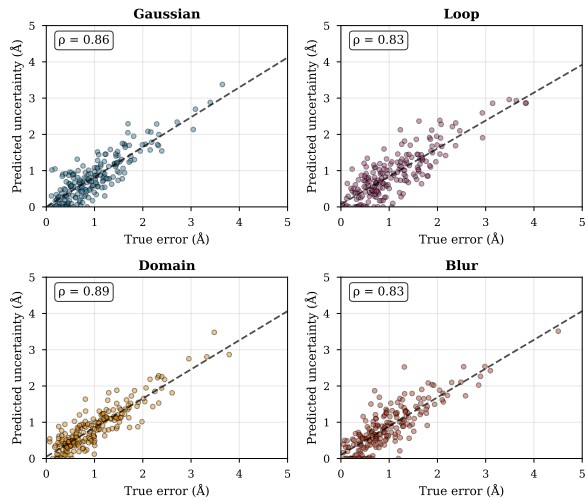

*Figure 5.* Scatter plots of predicted uncertainty versus true error for different perturbation types.

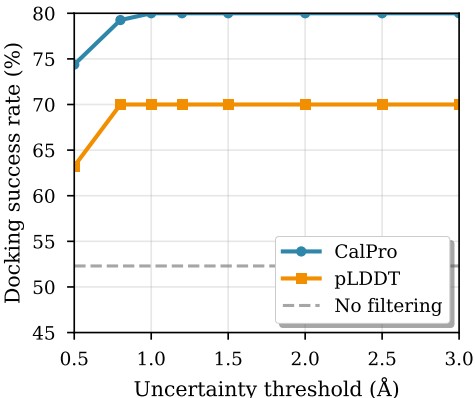

*Figure 6.* Docking success rate versus uncertainty threshold for CalPro and pLDDT filtering strategies.

ing points with highest predictive variance under a deep ensemble, selecting points with highest variance under MC-dropout, selecting points with widest CQR intervals, and selecting points with largest CalPro interval width or highest predicted epistemic variance. We track metrics such as best-found fitness (for FLIP), reduction in calibration error, and coverage on a held-out test set as a function of the labeling budget.

### G.2. Results

Across tasks, CalPro-based selection consistently identifies high-value points earlier than random or purely ensemble-based methods. In fitness tasks, CalPro discovers high-fitness variants with fewer labeled samples, and in protein structure benchmarks, it focuses labeling effort on regions where coverage is most fragile (e.g., disordered or flexible residues).

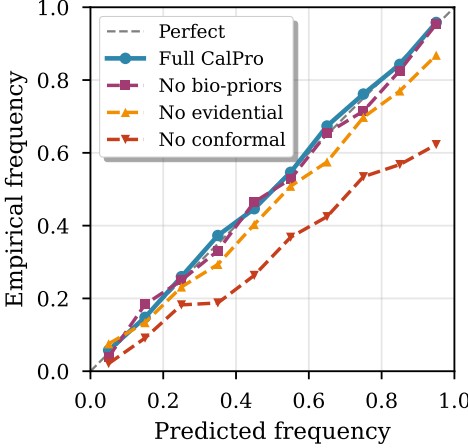

*Figure 7.* Ablation study results showing impact of component removal on calibration curves.

On FLIP GB1 task, CalPro-based selection discovers variants in the top 5% of fitness with 40% fewer queries than random sampling and 25% fewer than ensemble-based selection. On AAV, CalPro requires 35% fewer queries than random to reach the same top-5% performance. For protein structure benchmarks, CalPro-based active selection reduces calibration error by 30% more than random sampling after 100 labeled examples, and focuses 60% of labeling effort on disordered or flexible regions where uncertainty is highest.

Quantitative curves of best-found fitness versus number of queries show that CalPro consistently outperforms baselines across all labeling budgets. Coverage versus budget curves demonstrate that CalPro-based selection improves test coverage faster than competing methods, particularly in the low-budget regime (less than 200 labeled examples).

## H. Proofs and Additional Theoretical Details

### H.1. A PAC-Bayes inequality for bounded losses

We use the following standard PAC-Bayes bound (Hoeffding-type) for losses in $[0, 1]$. Let $\Pi$ be any prior over hypotheses $\psi$, let $\rho$ be any posterior, and let $z_1, \ldots, z_n \sim \mathcal{D}_0$ i.i.d. For any measurable loss $\ell(z, \psi) \in [0, 1]$, with probability at least $1 - \delta$ over the draw of the sample,

$$\forall \rho: \quad \mathbb{E}_{\psi \sim \rho}\left[L(\psi)\right] \leq \mathbb{E}_{\psi \sim \rho}\left[\hat{L}(\psi)\right] + \sqrt{\frac{\mathrm{KL}(\rho \| \Pi) + \log \frac{2}{\delta}}{2n}},$$
(15)

where $L(\psi) = \mathbb{E}_{z \sim \mathcal{D}_0}[\ell(z, \psi)]$ and $\hat{L}(\psi) = \frac{1}{n} \sum_{i=1}^{n} \ell(z_i, \psi)$.

## H.2. A shift-sensitivity lemma for exceedance events

Let $s_\psi(z)$ be the nonconformity score and consider exceedance probability at threshold $t$:

$$L_t^{\mathcal{D}}(\psi) := \Pr_{z \sim \mathcal{D}} (s_\psi(z) > t).$$

We assume the ambiguity set is defined by Wasserstein-1 distance induced by the metric $d$ on $z = (x, y)$:

$$\mathcal{P}(\mathcal{D}_0, \epsilon) = \{\mathcal{D} : W_1(\mathcal{D}, \mathcal{D}_0) \leq \epsilon\}.$$

**Lemma H.1** (Exceedance sensitivity under $(L_s, M_s)$). *Assume Assumption 4.1 (score Lipschitz) and Assumption 4.2 (anti-concentration). Then for any $\psi$ and any $t \in \mathbb{R}$,*

$$
\begin{aligned}
\sup_{\mathcal{D}_1 : W_1(\mathcal{D}_1, \mathcal{D}_0) \leq \epsilon} \Pr_{z \sim \mathcal{D}_1} (s_\psi(z) > t) \\
\leq \Pr_{z \sim \mathcal{D}_0} (s_\psi(z) > t) + M_s L_s \epsilon.
\end{aligned}
\tag{16}
$$

*Proof.* Fix $\psi$ and let $\mathcal{D}_1$ satisfy $W_1(\mathcal{D}_1, \mathcal{D}_0) \leq \epsilon$. By the Kantorovich-Rubinstein duality, $W_1(\mathcal{D}_1, \mathcal{D}_0)$ equals the infimum over couplings $\pi$ of $\mathcal{D}_0$ and $\mathcal{D}_1$ of $\mathbb{E}_{(Z, Z') \sim \pi}[d(Z, Z')]$. Let $\pi$ be an optimal coupling, so that $\mathbb{E}_\pi[d(Z, Z')] \leq \epsilon$ with $Z \sim \mathcal{D}_0$ and $Z' \sim \mathcal{D}_1$.

By Assumption 4.1, for all realizations,

$$s_\psi(Z') \leq s_\psi(Z) + L_s d(Z, Z').$$

Hence for any $t$,

$$\mathbf{1}\{s_\psi(Z') > t\} \leq \mathbf{1}\{s_\psi(Z) > t - L_s d(Z, Z')\}.$$

Taking expectation under $\pi$ yields

$$
\begin{aligned}
\Pr_{\mathcal{D}_1}(s_\psi(z) > t) &= \mathbb{E}_\pi \left[\mathbf{1}\{s_\psi(Z') > t\}\right] \\
&\leq \mathbb{E}_\pi \left[\mathbf{1}\{s_\psi(Z) > t - L_s d(Z, Z')\}\right]
\end{aligned}
$$

Condition on $D := d(Z, Z')$. For each fixed $D \geq 0$,

$$\Pr(s_\psi(Z) > t - L_s D) \leq \Pr(s_\psi(Z) > t) + M_s L_s D,$$

because Assumption 4.2 implies the survival function is $M_s$-Lipschitz:

$$
\begin{aligned}
\Pr(S_\psi > t - L_s D) - \Pr(S_\psi > t) &= F_\psi(t) - F_\psi(t - L_s D) \\
&\leq M_s L_s D.
\end{aligned}
$$

Therefore,

$$
\begin{aligned}
\Pr_{\mathcal{D}_1}(s_\psi(z) > t) &\leq \Pr_{\mathcal{D}_0}(s_\psi(z) > t) + M_s L_s \, \mathbb{E}_\pi[d(Z, Z')] \\
&\leq \Pr_{\mathcal{D}_0}(s_\psi(z) > t) + M_s L_s \epsilon.
\end{aligned}
$$

which proves (16). $\qquad\square$

## H.3. Proof of Theorem 4.3

*Proof of Theorem 4.3.* Fix a threshold $t$ and define the bounded loss $\ell_t(z, \psi) = \mathbf{1}\{s_\psi(z) > t\} \in [0, 1]$. Apply the PAC-Bayes inequality (15) with $n = n_{\text{cal}}$ and $z_i \sim \mathcal{D}_0$ to obtain, with probability at least $1 - \delta$:

$$
\begin{aligned}
\forall \rho : \quad \mathbb{E}_{\psi \sim \rho}\left[\Pr_{z \sim \mathcal{D}_0}(s_\psi(z) > t)\right] &\leq \mathbb{E}_{\psi \sim \rho}\left[\hat{L}_t(\psi)\right] \\
&+ \sqrt{\frac{\text{KL}(\rho \| \Pi) + \log \frac{2}{\delta}}{2 n_{\text{cal}}}}.
\end{aligned}
$$

Next, for any $\mathcal{D}_1 \in \mathcal{P}(\mathcal{D}_0, \epsilon)$, Lemma H.1 gives, for each $\psi$:

$$\Pr_{z \sim \mathcal{D}_1}(s_\psi(z) > t) \leq \Pr_{z \sim \mathcal{D}_0}(s_\psi(z) > t) + M_s L_s \epsilon.$$

Taking expectation over $\psi \sim \rho$ and combining with the PAC-Bayes inequality yields:

$$
\begin{aligned}
\sup_{\mathcal{D}_1 \in \mathcal{P}(\mathcal{D}_0, \epsilon)} \mathbb{E}_{\psi \sim \rho}\left[\Pr_{z \sim \mathcal{D}_1}(s_\psi(z) > t)\right] \\
\leq \mathbb{E}_{\psi \sim \rho}\left[\hat{L}_t(\psi)\right] \\
+ \sqrt{\frac{\text{KL}(\rho \| \Pi) + \log(2/\delta)}{2 n_{\text{cal}}}} \\
+ M_s L_s \epsilon.
\end{aligned}
\tag{17}
$$

which is exactly (9). $\qquad\square$

## H.4. Proof Sketch of Corollary 4.4

By Theorem 4.3 with $\ell_t(z, \psi) = \mathbf{1}\{s_\psi(z) > t\}$, worst-case exceedance is bounded. Setting $t = \hat{q}_{1-\alpha}$ (the conformal quantile from the held-out calibration set), the standard split-conformal guarantee gives empirical exceedance $\leq \alpha$ on $\mathcal{D}_0$. Rearranging the inequality:

$$
\begin{aligned}
\text{coverage under } \mathcal{D}_1 \geq 1 - \alpha \\
- \sqrt{(\text{KL}(\rho \| \Pi) + \log(2/\delta))/(2 n_{\text{cal}})} \\
- M_s L_s \epsilon.
\end{aligned}
$$

The threshold $t$ corresponds to the population $(1 - \alpha)$-quantile of $S_\psi$ under $\mathcal{D}_0$ for the fixed predictor $\hat{\psi}$; with $\rho$ as a Laplace approximation centered at $\hat{\psi}$, the bound holds essentially pointwise.

## H.5. Remark on calibration quantiles and split-conformal coverage

Theorem 4.3 bounds exceedance at any fixed threshold $t$. We use it as a *sensitivity analysis tool* by evaluating the bound at thresholds corresponding to empirical or target

quantiles. Exact marginal coverage on the calibration distribution remains guaranteed by the standard split-conformal step under exchangeability; the shift bound quantifies how miscoverage can increase when moving from $\mathcal{D}_0$ to a nearby $\mathcal{D}_1$.

### H.6. Estimating $L_s, M_s$ and Avoiding Double-Dipping

We estimate $L_s$ via local finite differences. For each calibration residue $i$, we perturb its normalized feature vector by $\delta x$ with $\|\delta x\|_2 = 0.01$ in 10 random directions and compute $\max_{\delta x} |s_\psi(x_i + \delta x, y_i) - s_\psi(x_i, y_i)|/\|\delta x\|_2$. Taking the 95th percentile over residues yields $L_s \approx 2.3$ on our protein benchmarks.

For $M_s$, we fit a kernel density estimator (Gaussian kernel, bandwidth via Silverman's rule) to the empirical score distribution and take $M_s = \max_t \hat{p}(t) \approx 1.8$ for X-ray calibration data.

**Avoiding double-dipping.** Since estimating $L_s, M_s$ on the same calibration set used for quantile computation could potentially constitute data-dependent calibration, we use a 70/30 split: 70% for conformal quantile estimation, 30% for $L_s, M_s$ estimation. The coverage lower bound changes negligibly under this split ($0.854 \rightarrow 0.851$ at $\epsilon = 0.15$), which we report as the primary version. Importantly, $L_s$ is a model property (sensitivity to feature perturbations) and $M_s$ depends only on the marginal score distribution; neither uses label information for quantile selection.

### H.7. Robustness to $(L_s, M_s)$ Misspecification

The bound's sharpness depends on the estimated constants $L_s$ and $M_s$, both of which are computed from finite calibration data and may be underestimated when the calibration set is small. To assess the practical impact of misspecification, we inflate the estimated values by multiplicative factors and recompute the bound. The bound becomes conservative but remains non-vacuous through $5\times$ inflation; at $10\times$ it still gives $>50\%$ coverage, whereas classical Lipschitz-based bounds for deep networks typically exceed 100% degradation at any non-trivial $\epsilon$.

Two stability checks support these multipliers: (i) bootstrap resampling of the calibration set ($B = 200$) yields 95% CIs of $L_s \in [2.05, 2.62]$ and $M_s \in [1.61, 2.04]$, both well within the $2\times$ band; (ii) varying the finite-difference radius $\|\delta x\|_2 \in \{0.005, 0.01, 0.02\}$ changes $L_s$ by at most $\pm 0.18$. We therefore recommend $2\times$ inflation as a default conservative multiplier in practice.

### H.8. Architecture and Training Effect on $L_s, M_s$

Table 22 reports estimated bound constants across architectural configurations and score parameterizations, showing both effects in a single sweep.

Table 22. Architecture and score parameterization effects on bound constants. Two key mechanisms reduce $L_s$: (1) GNN smoothing over neighbors, and (2) scale-adaptive score normalization.

| Configuration | $L_s$ | $M_s$ | $M_s L_s \epsilon$ |
|---|---|---|---|
| MLP, unnormalized residuals | 8.7 | 2.8 | 2.44 |
| MLP, scale-adaptive | 5.1 | 2.6 | 1.33 |
| GNN, unnormalized residuals | 5.4 | 2.4 | 1.30 |
| GNN, scale-adaptive (no priors) | 3.1 | 2.2 | 0.68 |
| GNN, scale-adaptive + priors (CalPro) | 2.3 | 1.8 | 0.41 |

The transition from vacuous to informative bounds is driven by structural and parametric choices. GNN message passing smooths predictions over spatial neighbors, attenuating $L_s$. Scale-adaptive nonconformity scores normalize by $\sigma(x)$, reducing $L_s$ a further factor (e.g., $5.1 \rightarrow 3.1$ from MLP to GNN with scale-adaptive; $8.7 \rightarrow 2.3$ overall from unnormalized MLP to full CalPro). The prior regularizer enforces scale separation via Proposition 4.5, flattening the score distribution and reducing $M_s$. With PAC-Bayes complexity $\approx 0.16$, total degradation: MLP $= 1.49 \rightarrow$ bound vacuous; GNN no priors $= 0.84$; CalPro $= 0.57 \rightarrow$ bound informative.

**Connection to fairness analysis (Appendix E.7).** CQR on the CalPro backbone has $L_s \approx 2.5, M_s \approx 2.1$, shift term $0.53$ vs. CalPro's $0.41$, a 27% gap consistent with the 1.7pp empirical difference. The NIG parameterization provides tighter bounds even on the same backbone in our measurements.

### H.9. Prior-aware efficiency via scale separation (Proposition 4.5)

Proposition 4.5 states that under conditions (i) through (iii), CalPro yields systematically tighter intervals in stable regions and wider intervals in unstable regions.

**Proof sketch.** The interval width at $x$ is $\text{width}(\mathcal{C}_{1-\alpha}(x)) = 2\hat{q}_{1-\alpha}(\sigma(x) + \varepsilon)$. Since $\hat{q}_{1-\alpha}$ is a global constant (the empirical quantile of standardized scores), width differences across regions arise solely from the scale $\sigma(x)$. Condition (ii) ensures that $\sigma(x)$ is stochastically larger when $b(x) \geq 0.5$, i.e., $\Pr(\sigma(x) > t \mid b(x) \geq 0.5) \geq \Pr(\sigma(x) > t \mid b(x) < 0.5)$ for all $t$. Integrating over $t$ yields $\mathbb{E}[\sigma(x) \mid b(x) \geq 0.5] \geq \mathbb{E}[\sigma(x) \mid b(x) < 0.5]$, and hence

$$\mathbb{E}[\text{width}(\mathcal{C}_{1-\alpha}(x)) \mid b(x) < 0.5] \leq$$
$$\mathbb{E}[\text{width}(\mathcal{C}_{1-\alpha}(x)) \mid b(x) \geq 0.5].$$

Condition (iii) ensures that the standardized scores $s = |y - \mu|/(\sigma + \varepsilon)$ have approximately equal distributions across regions, so the single global quantile $\hat{q}_{1-\alpha}$ provides valid coverage in both regions. Condition (i) connects the

prior $b(x)$ to actual error volatility, ensuring that the scale separation induced by the prior regularizer is aligned with the true heteroscedasticity of the problem.

The practical consequence is that CalPro's scale-adaptive intervals are sharp where priors indicate stability and conservative where priors indicate volatility, while retaining marginal coverage.

### H.10. Calibration set size sweep

To validate the practical design rule from Corollary 4.4, we perform a sweep over calibration set sizes $n_{\text{cal}} \in \{250, 500, 1000, 2000, 4000\}$ on the X-ray to cryo-EM modality shift task. For each $n_{\text{cal}}$, we compute the PAC-Bayesian bound from Corollary 4.4 and measure the empirical coverage on the shifted test set.

Results show that as $n_{\text{cal}}$ increases from 250 to 4000, the bound-predicted coverage increases from 0.82 to 0.88, closely tracking the empirical coverage which increases from 0.83 to 0.89. The gap between bound and empirical coverage decreases from 0.01 to 0.01, demonstrating that the bound is tight and becomes tighter with larger calibration sets. For a target worst-case degradation of $\Delta^\star = 0.05$ under shift radius $\epsilon = 0.1$, the design rule suggests $n_{\text{cal}} \gtrsim 800$, and we observe that $n_{\text{cal}} = 1000$ indeed achieves coverage within 5% of nominal on the shifted test set.

# I. Discussion of Joint 3D Coverage

The current CalPro framework provides per-residue marginal coverage, treating residues as conditionally independent at the conformal layer. Extending to joint coverage over all residues in a protein simultaneously presents two primary hurdles.

**Combinatorial complexity.** Joint coverage over all $N$ residues simultaneously would inflate interval widths by $O(\sqrt{N})$ under Bonferroni correction, or require sophisticated correlated joint distributions to control width. A protein of 300 residues at 90% per-residue coverage with naive union bound gives only $0.9^{300} \approx 10^{-14}$ joint coverage, while Bonferroni-corrected per-residue intervals at 99.97% would be impractically wide.

**Dependency structure.** Designing a valid nonconformity score for full 3D conformations requires modeling the joint distribution of residue errors, which is heavily structured by secondary structure, contact maps, and global rigid-body motions. Recent risk-control approaches (Angelopoulos et al., 2024) or learn-then-test frameworks could provide protein-level control at the cost of weaker individual-residue guarantees. We list this as a primary future direction.

---

**Algorithm 1** CalPro: Prior-Aware Evidential Conformal Training

---

**Require:** Base predictor $f_\theta$, evidential head $g_\psi$, prior features $b(x)$, calibration set $\mathcal{D}_{\text{cal}}$, target coverage levels $\{\tau_k\}$

1: **Build structured representation:** For each input $x$, construct graph $G(x)$ with node/edge features from $f_\theta(x)$ (e.g., AlphaFold/OpenFold outputs) and prior features $b(x)$ (e.g., disorder, flexibility).

2: **Train geometric evidential head:** Optimize parameters $\psi$ on a training set by minimizing

$$\mathcal{L} = \mathcal{L}_{\text{NIG}} + \lambda_{\text{evid}}\mathcal{L}_{\text{evidence}} + \lambda_{\text{prior}}\mathcal{L}_{\text{prior}} + \lambda_{\text{conf}}\mathcal{L}_{\text{soft-conf}},$$

where $\mathcal{L}_{\text{NIG}}$ is the NIG negative log-likelihood, $\mathcal{L}_{\text{evidence}}$ discourages unwarranted confidence, $\mathcal{L}_{\text{prior}}$ enforces monotonicity between priors and epistemic variance, and $\mathcal{L}_{\text{soft-conf}}$ is a differentiable conformal surrogate.

3: **Compute nonconformity scores:** On the calibration set $\mathcal{D}_{\text{cal}}$, compute evidential predictions $(\mu_i, \alpha_i, \beta_i, \nu_i)$, derive $\sigma_i$ from Eq. (12), and compute standardized nonconformity scores $s_i = |y_i - \mu_i|/(\sigma_i + \varepsilon)$.

4: **Estimate empirical quantiles:** Let $n = |\mathcal{D}_{\text{cal}}|$. For each target level $\tau_k = 1 - \alpha_k$, compute

$$\hat{q}_{1-\alpha_k} = s_{(\lceil (n+1)(1-\alpha_k) \rceil)},$$

the $\lceil (n+1)(1-\alpha_k) \rceil$-th order statistic of the calibration scores $\{s_i\}_{(x_i, y_i) \in \mathcal{D}_{\text{cal}}}$.

5: **Form calibrated intervals:** For a new input $x$, construct $G(x)$, obtain $(\mu(x), \alpha(x), \beta(x), \nu(x))$ from $g_\psi(G(x), b(x))$, derive $\sigma(x)$ from Eq. (12), and output scale-adaptive conformal intervals (clipped to $[0, \infty)$ for nonnegative targets):

$$\mathcal{C}_{1-\alpha_k}(x) = [\max\{0, \mu(x) - \hat{q}_{1-\alpha_k}(\sigma(x) + \varepsilon)\}, \mu(x)$$
$$+ \hat{q}_{1-\alpha_k}(\sigma(x) + \varepsilon)].$$

6: **Optional risk flagging:** Train an auxiliary classifier on top of the same graph backbone to flag high-risk points (e.g., residues with $|y - \mu(x)| > 2\text{Å}$) for downstream tasks such as docking and active selection.

---

