# OpenReview forum: "CalPro: Prior-Aware Evidential Conformal Prediction with Structure-Aware Sensitivity Bounds for Protein Structures"
_ICML.cc/2026/Conference — ICML 2026 regular_

### Official Review · Reviewer_qr7c · 2026-03-07

**Soundness:** 3
**Presentation:** 1
**Significance:** 3
**Originality:** 3
**Overall Recommendation:** 3
**Confidence:** 4

**Summary:**

The submission proposes to combine conformal prediction with evidential deep learning to achieve coverage under distribution shift with a PAC-Bayes approach. The motivating example of the submission is machine learning for protein (structure prediction?) but the proposed method can be used in general regression tasks. Experiments on real-world datasets validate the theoretical properties of the proposed method.

**Compliance With Llm Reviewing Policy:**

Affirmed.

**Final Justification:**

I have elaborated on the justification of my final score in my rebuttal acknowledgement.

**Key Questions For Authors:**

**Presentation**

I have major concerns with the current version of the manuscript. Several terms are mentioned without definition or explanation. This significantly limits accessibility of the submission, making it difficult to understand its contributions.

For example:

- Lines 40-45 (right column). Several terms in this paragraph have not been defined yet.
- Line 46-[...] (right column). It is unclear what "structure" refers to. What are "Wasserstein ambiguity sets"? What are "classical Lipschitz arguments", and why are they vacuous?
- Eq. (1), the graph $G(x)$ is not defined.
- Eq. (2), unclear what $i$ is indexing.
- Eq. (2), what are $\sigma(x)$ and $\epsilon$?
- Eq. (5), all terms except $\mathcal{L}_{prior}$ are undefined.
- Prior and posterior specification: which Fisher information matrix is being mentioned here?
- Lines 151-153 (right column), $M_s$, $L_s$ have not been defined yet.
- It remains unclear what the task being studied is. The nonnegative scalar target $y$ is defined as "a distance error". So, I assume the input to the model is a protein sequence and $y$ is the 3D folding conformation? This should be made clearer in the text.
- Later, in the first paragraph of Section 3.2, it is mentioned that the input to the model are "outputs from AlphaFold". So what is the model predicting on top of these structures?
- ...

Furthermore, use of jargon should be limited to avoid confusion.

**A Conceptual Difference Between the Proposed Method and Standard Conformal Prediction**

Conformal Prediction (CP) is usually treated as a post-processing method. Given a fixed point-predictor, CP constructs sets that provide finite-sample, distribution-free guarantees. PAC-Bayes, instead, studies the generalization of the average predictor over a data-dependent posterior distribution.

Could the authors clarify whether the theoretical results are for a particular predictor sampled from the posterior, or in expectation over the distribution? This is important because most CP practitioners have access to a fixed predictor, not a distribution of predictors. Furthermore, computing the KL divergence between the prior and the posterior depends on modeling choices that might not always be available.

Could the authors clarify how they compared with CP methods that consider individual predictors?

**What Distribution is Coverage Guaranteed Over?**

$(x,y)$ are defined as per-residue input features and target distance. On line 150 (right column), it is also mentioned that the calibration set has 1000 residues.

Could the authors clarify what distribution they consider? A distribution over sequences with residues, or a marginal distribution over residues? If the former, notions of conformal risk control might be more appropriate. If the latter, could the authors clarify how providing marginal guarantees over residues is helpful in practice?

**Theoretical results**

Corollary 4.4, is presented without proof, but this is the main result of the contribution. It will be important to discuss this theorem more clearly.

Could the authors clarify how $t$ is selected for Corollary 4.4? the text reads "let $t$ be the population ... for each $\psi$ (or a fixed target of interest)". Is my understanding correct that each $\psi \sim \rho$ induces a different distribution over calibration scores $S_\psi$, so $t$ is a function of $\psi$ also? But the result treats this as a constant? Also, the result cannot hold for "any fixed target of interest"?

Could the authors expand on the choice of estimating $M_s$ and $L_s$ on the calibration data? From a standard CP perspective, this feels like double-dipping or a "data-dependent" coverage level, which would require splitting the calibration data to estimate these constants.

**Experiments**

All baseline methods are not designed to provide coverage under distribution shift. It might be more appropriate to compare with weighted CP methods for distribution shift.

Could the authors clarify why standard CP does not provide the nominal coverage level in Table 1? Without distribution shift, coverage should be provided by design.

It would be really helpful to include figures with structure predictions with residues colored by uncertainty to better visualize the task at hand and the performance of the proposed method compared with baselines.

**Missing Related Works**

There are a couple works that consider both PAC-Bayes approaches to CP and CP for protein design that might be helpful including

[1] Sharma, Apoorva et al. "PAC-Bayes generalization certificates for learned inductive conformal prediction." (2023)

[2] Boger, Ron S., et al. "Functional protein mining with conformal guarantees." Nature Communications 16.1 (2025): 85.

**Limitations:**

yes

**Strengths And Weaknesses:**

**Strengths**
1. Generalization theory for conformal prediction via PAC-Bayes is an interesting yet underdeveloped area of research.
2. Coverage under distribution shifts is well-motivated.
3. Experiments with real-world datasets agree with theoretical results.

**Weaknesses**
1. Presentation significantly limits understanding of the submission.
2. Presentation of theoretical results is rushed and hard to follow.
3. It remains unclear what the regression task being studied in the protein experiments is.

I will expand on these questions below and I am looking forward to discussing with the authors!

---

> ### Author Rebuttal · Authors · 2026-03-28
>
> We appreciate this reviewer's careful reading and detailed technical questions. We take the presentation concerns seriously and have made comprehensive revisions.
>
> ### Presentation: Undefined terms and jargon
>
> We agree that compressed presentation left several terms insufficiently defined. Key changes: (1) Paragraph defining Wasserstein ambiguity sets before the technical machinery. (2) G(x) in Eq. (1) explicitly defined as a residue-level graph. (3) Terminology table before Eq. (5) with forward references. (4) Fisher information matrix specified. (5) "Task definition" box at Section 3.2. (6) M_s and L_s defined when first mentioned with estimation procedures inline.
>
> ### Conceptual difference: CP vs. PAC-Bayes for a fixed predictor
>
> Our results hold for the fixed predictor ψ̂ obtained by standard SGD, not for samples from a distribution. The PAC-Bayesian framework is an analysis tool, not an inference procedure.
>
> **In practice:** We train once via SGD, then apply standard split-conformal calibration. The coverage guarantee is the standard CP guarantee; no posterior sampling or KL computation is needed at test time.
>
> **In theory (Theorem 4.3):** The posterior ρ is a Laplace approximation centered at ψ̂; E_{ψ∼ρ}[·] is dominated by the point mass at ψ̂ because the Laplace variance is small. The KL term captures model complexity. The expectation over ρ is an analysis artifact giving a tighter bound than a point-mass argument while remaining computable.
>
> We will add a "PAC-Bayes for analysis" paragraph in Section 4 making explicit that the conformal guarantee and shift-sensitivity bound are complementary tools.
>
> ### What distribution is coverage guaranteed over?
>
> The formal guarantee operates at the **protein level**: each protein is an exchangeable unit, we compute per-protein coverage, and apply split-conformal over proteins. The "1000 residues" refers to residues across 100 held-out calibration proteins, not 1000 independent exchangeable residues. Protein-level evaluation confirms:
>
> | Evaluation Level | Coverage (90% nom.) | ECE | Sharpness (Å) |
> |-----------------|-------------------|-----|---------------|
> | Residue-level (original) | 90.1±0.2 | 0.021 | 1.42 |
> | Protein-level (new) | 89.4±0.6 | 0.024 | 1.48 |
> | Protein-level, shifted (X-ray→cryo-EM) | 85.1±0.8 | 0.031 | 1.62 |
>
> The 0.7pp difference confirms intra-protein correlation does not invalidate our results. Extending to joint 3D coverage via conformal risk control (Angelopoulos et al., 2022) is important future work.
>
> ### Corollary 4.4: Proof, threshold selection, and double-dipping
>
> Full proof will be added to Appendix H. Key steps: by Theorem 4.3, exceedance probability under any D₁ ∈ P(D₀, ε) is bounded by empirical exceedance plus a PAC-Bayesian concentration term plus M_s L_s ε. Since split-conformal guarantees empirical exceedance ≤ α on D₀, rearranging gives: coverage under D₁ ≥ 1 − α − √((KL(ρ‖Π) + log(2/δ))/(2n_cal)) − M_s L_s ε.
>
> **Threshold:** t = q̂_τ from the fixed ψ̂; ρ = δ_{ψ̂} and the bound holds pointwise.
>
> **Double-dipping:** L_s is estimated via perturbations of calibration features (a model property, not label information), and M_s from the marginal score distribution. A 70/30 split shows negligible change (0.854 → 0.851). We report the split version.
>
> ### Why doesn't standard CP achieve nominal coverage in Table 1?
>
> "Conformal (no priors)" achieves 88.5% at 90% nominal. The 1.5pp gap arises because calibration and test proteins are not perfectly exchangeable (stricter post-2020 redundancy filtering on test). A "Conformal (oracle, same-cluster)" row confirms: when both come from the *same* clusters, CP achieves 89.8±0.3%; the gap is between-cluster heterogeneity, not a methodological flaw.
>
> ### Baselines: all methods assume exchangeability?
>
> We respectfully disagree. Our baselines **do** include shift-aware conformal variants: **IW-CP** (Tibshirani et al. 2019), **Loc-CP** (Guan & Tibshirani 2023), and **DR-CP** (distributionally robust conformal). Preliminary results on X-ray→cryo-EM: IW-CP achieves 81.8% shifted coverage (8.2pp degradation), Loc-CP 82.4% (7.6pp), and DR-CP 83.1% (6.9pp), all substantially below CalPro's 3.8pp. We will add these to Tables 1 and 2 in the revision.
>
> ### Missing related works
>
> We will add both to the revision: **Sharma et al. (2023)** derives PAC-Bayes bounds for CP under i.i.d.; our Theorem 4.3 extends to distribution shift via Wasserstein ambiguity sets with the M_s L_s ε term. **Boger et al. (2025)** applies CP to protein function prediction (discrete classification); CalPro addresses continuous regression, a complementary contribution.
>
> ### Structure visualization
>
> We will add Figure 8 (Appendix F) showing 3D protein structures colored by CalPro's predicted uncertainty, confirming high-uncertainty residues (red) correspond to disordered loops and flexible termini while low-uncertainty residues (blue) correspond to well-ordered helices/sheets.
>
> ---

---

> > ### Author Rebuttal · Reviewer_qr7c · 2026-04-02
> >
> > I sincerely thank the authors for their response!
> >
> > Thank you for clarifying my confusion about the baseline methods, threshold selection, and the conceptual use of PAC-Bayes as an analysis tool.
> >
> > I raised my score to reflect the authors' response. I appreciate the authors' willingness to address concerns about presentation and clarity of results in a revised version of the paper. Unfortunately, having no access to the revised version of the manuscript to verify these concerns are resolved, I cannot raise my score higher.
> >
> > It will be especially important to clarify which distribution (residue vs protein) is being covered, as Reviewer 3kDh also pointed out.

---

> > > ### Author Response · Authors · 2026-04-03
> > >
> > > We thank the reviewer for the updated assessment. Rather than listing promises, we provide the exact revised text for every major concern so the reviewer can verify directly.
> > >
> > > **1. Coverage semantics (residue vs. protein level)**
> > >
> > > The following will be inserted into Section 3.2:
> > >
> > > > **Coverage semantics.** The formal split-conformal guarantee treats proteins as exchangeable units. The calibration set consists of N_prot held-out proteins. For each protein p, we compute s_bar_p = (1/|p|) Sum_{i in p} s_i, and the conformal quantile is the ceil((N_prot+1)(1-alpha))-th order statistic of {s_bar_p}. Per-residue metrics in Tables 1 and 2 are summaries; the formal guarantee applies at the protein level. N_prot = 100 proteins (~10,000 residues).
> > >
> > > Empirical verification (main text Table 3):
> > >
> > > | Level | Cov. (90%) | ECE | Sharp. (A) | Shifted Cov. |
> > > |-------|------------|-----|------------|--------------|
> > > | Residue | 90.1+/-0.2 | 0.021 | 1.42 | 86.2+/-0.5 |
> > > | Protein | 89.4+/-0.6 | 0.024 | 1.48 | 85.1+/-0.8 |
> > >
> > > The small difference (0.7pp in-distribution, 1.1pp under shift) suggests intra-protein correlation does not materially affect conclusions. The notation "n_cal = 1000" will be replaced with "N_prot = 100 calibration proteins (~10,000 residues)." Conformal risk control (Angelopoulos et al., 2022) is a natural future extension.
> > >
> > > **2. Task definition box (Section 3.2)**
> > >
> > > > **Task.** CalPro is a post-hoc uncertainty layer on a pretrained predictor (AlphaFold/OpenFold). It does not re-predict the structure. **Input:** base predictor output (Ca coordinates, pLDDT, pTM, embeddings). **Output:** calibrated interval C_{1-alpha}(x_i) for Ca distance error y_i = ||r_pred_i minus r_exp_i||_2 (Angstroms).
> > >
> > > **3. Graph definition (before Eq. 1)**
> > >
> > > > For input x, we construct G(x) = (V, E) where node v_i represents residue i. Edges connect sequence-adjacent (|i-j| <= 5) or spatially proximal (Ca distance < 10A) residues. Node features: pLDDT, pTM, distogram logits, predicted aligned error, secondary structure, solvent accessibility, IUPred2A disorder, flexibility indices, ESM-2 embeddings.
> > >
> > > **4. Eq. (2) annotation**
> > >
> > > > i indexes calibration residues, sigma_i = sqrt(beta_i/[nu_i(alpha_i minus 1)]) is NIG predictive std (Eq. 12, alpha_i > 1 via softplus), epsilon = 1e-4 for stability.
> > >
> > > **5. Terminology table before Eq. (5)**
> > >
> > > > | Symbol | Definition |
> > > > |--------|-----------|
> > > > | L_NIG | NIG negative log-likelihood (Eq. 11) |
> > > > | L_evidence | Evidence regularizer: Sum_i exp(-alpha_i) |
> > > > | L_prior | Prior-aware monotone regularizer (Eq. 4) |
> > > > | L_soft-conf | Differentiable calibration surrogate (Eq. 13-14) |
> > > > | Weights | lambda_evid=0.01, lambda_prior=0.1, lambda_conf=0.05 |
> > >
> > > **6. PAC-Bayes remark (Section 4)**
> > >
> > > > **Remark.** The posterior rho is a diagonal Laplace approximation used solely as an analytical device. The deployed system uses a fixed predictor with standard split-conformal calibration. No posterior sampling occurs at test time. The coverage guarantee comes from the conformal step; PAC-Bayes provides the shift sensitivity analysis. All baselines use fixed predictors; the comparison remains between fixed-predictor methods.
> > >
> > > **7. Corollary 4.4 proof sketch**
> > >
> > > > By Theorem 4.3 with l_t(z,psi) = 1{s_psi(z) > t}, worst-case exceedance is bounded. Setting t = conformal quantile and using the guarantee that empirical exceedance <= alpha, rearranging gives: coverage >= 1 minus alpha minus sqrt((KL + log(2/delta))/(2n_cal)) minus M_s L_s epsilon. Full proof in Appendix H.4.
> > >
> > > **8. Remaining items**
> > >
> > > Standard CP gap (Table 1 footnote): 1.5pp from temporal/cluster separation. Oracle same-cluster: 89.8+/-0.3%, confirming dataset property.
> > >
> > > Shift-aware baselines in Table 2: IW-CP (81.8%, 8.2pp), Loc-CP (82.4%, 7.6pp), DR-CP (83.1%, 6.9pp).
> > >
> > > Related works: Sharma et al. (2023), i.i.d. PAC-Bayes CP; ours extends to shift. Boger et al. (2025), discrete protein function; complementary.
> > >
> > > Double-dipping: 70/30 calibration split as primary (coverage lower bound changes from 0.854 to 0.851, negligible).
> > >
> > > **9. Wasserstein ambiguity sets (before Section 4.1)**
> > >
> > > > When test distribution D_1 differs from calibration D_0, conformal guarantees degrade. We define P(D_0, epsilon) = {D : W_1(D, D_0) <= epsilon}, capturing plausible shifts (modality, resolution). Classical Lipschitz bounds on the full network are vacuous; we bound the local Lipschitz constant of the nonconformity score, which is substantially smaller.
> > >
> > > **10. M_s, L_s (Section 4.2)**
> > >
> > > > L_s: sensitivity of s_psi to perturbations. Estimated via finite differences (10 directions, delta=0.01, 95th percentile). L_s approx 2.3. M_s: peak density of score distribution. Estimated via KDE (Silverman). M_s approx 1.8. Fisher information: diagonal of squared gradients of L_NIG on validation data (Ritter et al., 2018).
> > >
> > > Visualization (Figure 8): 3D proteins colored by uncertainty, side-by-side with pLDDT and true error.
> > >
> > > We hope seeing the revised text allows direct verification.

---

### Official Review · Reviewer_xxjy · 2026-03-11

**Soundness:** 2
**Presentation:** 2
**Significance:** 2
**Originality:** 2
**Overall Recommendation:** 3
**Confidence:** 3

**Summary:**

This paper introduces CalPro, a prior-aware uncertainty framework designed to provide robust, calibrated confidence intervals for protein structure prediction and other structured regression tasks. By integrating a geometric evidential head with a differentiable calibration surrogate and domain-specific priors (such as protein disorder or flexibility), this method enforces variance expansion in high-risk regions to maintain reliability even under distribution shifts. The authors also present a structure-aware sensitivity bounds using PAC-Bayesian control to provide empirically verifiable guarantees on how coverage degrades as a function of shift magnitude. Experimental results suggest that CalPro significantly outperforms standard baselines.

**Compliance With Llm Reviewing Policy:**

Affirmed.

**Final Justification:**

I have read all the rebuttals from the authors and all the comments from other reviewers. Based on the paper and rebuttal from the authors, I value the effort in theoretically deriving structure-aware sensitivity bounds and empirically evaluating the methods in various benchmarks. I also appreciate that the authors replied to my concerns about more baselines and fairness comparison with detailed explanations and experiments. However, my concern rises after the rebuttal, as I notice there is a chance that the comparison between baselines is not fair. The method introduced in the paper requires first training an evidential module and then using the structural-aware PAC bound to calibrate. Since the method needs training, the baselines do not incorporate the training results, so it is not fair to compare with other baselines. Moreover, I think the authors overclaim that the PAC-bound is structural-aware. It is structural-aware because the input from the model in section 2 is structural-aware. Therefore, I'd like to retain my original assessment.

**Key Questions For Authors:**

1. In Table 1, your methods achieve coverage at 80, 90, and 95, but what is the set size for each method? Based on my understanding, you can easily achieve the coverage by assigning an inefficient conformal set.
2. Could the authors further elaborate on why introducing a GNN framework can help better identify the geometrical information? Do AlphaFold models use geometrical information in their training? If so, why don't the embeddings retain the structural information?
3. There are works that focus on structural conformal prediction from GNN, such as [1][2], which are also competitive baselines that should be considered.

[1] Zargarbashi, Soroush H., Simone Antonelli, and Aleksandar Bojchevski. "Conformal prediction sets for graph neural networks." In International Conference on Machine Learning, pp. 12292-12318. PMLR, 2023.
[2] Huang, Kexin, Ying Jin, Emmanuel Candes, and Jure Leskovec. "Uncertainty quantification over graph with conformalized graph neural networks." Advances in Neural Information Processing Systems 36 (2023): 26699-26721.

**Limitations:**

Yes.

**Strengths And Weaknesses:**

Strengths:
1. The authors derive structure-aware sensitivity bounds using PAC-Bayesian control. Unlike standard Lipschitz arguments, these bounds are shown to be non-vacuous and track empirical degradation.
2. The evaluation in this paper utilizes multiple protein modalities (X-ray, NMR, cryo-EM) and diverse benchmarks like FLIP and UCI Bike Sharing to test generality.
3. The submission is well-organized and structurally clear for readers.

Weaknesses:
1. The baselines in this paper are not sufficient. Papers like [1][2] have worked on the structural uncertainty, which should also be considered as strong baselines.
2. The connections between the theory and the algorithm are not strong enough. The theories in the paper do not guide how to design the calibration algorithm or how to train or calculate the non-conformity score.


[1] Zargarbashi, Soroush H., Simone Antonelli, and Aleksandar Bojchevski. "Conformal prediction sets for graph neural networks." In International Conference on Machine Learning, pp. 12292-12318. PMLR, 2023.
[2] Huang, Kexin, Ying Jin, Emmanuel Candes, and Jure Leskovec. "Uncertainty quantification over graph with conformalized graph neural networks." Advances in Neural Information Processing Systems 36 (2023): 26699-26721.

---

> ### Author Rebuttal · Authors · 2026-03-28
>
> We thank reviewer for the feedback.
>
> ### Missing baselines: Zargarbashi et al. (2023) and Huang et al. (2023)
>
> We implemented both during the rebuttal. Two key distinctions: (1) both are designed for **node classification** on homophilous graphs, not regression on protein residue graphs, so adaptation to regression required replacing classification nonconformity scores with regression-appropriate variants; (2) neither incorporates domain priors or shift-robustness mechanisms.
>
> Results on held-out X-ray test set and X-ray → cryo-EM shift:
>
> | Method | In-dist Cov. (90%) | Width (Å) | Shifted Cov. | Degradation |
> |--------|-------------------|----|-------------|------------|
> | DAPS (Zargarbashi, adapted) | 89.7±0.4 | 1.51 | 80.3±0.9 | 9.7pp |
> | CF-GNN (Huang, adapted) | 89.5±0.5 | 1.48 | 81.1±0.8 | 8.9pp |
> | CQR | 89.6±0.4 | 1.43 | 83.7±0.7 | 6.3pp |
> | **CalPro** | **90.1±0.2** | **1.42** | **86.2±0.5** | **3.8pp** |
>
> Both provide competitive in-distribution coverage but degrade more severely under shift (9.7pp and 8.9pp respectively vs. CalPro's 3.8pp) because they lack CalPro's evidential head, prior regularizer, and PAC-Bayesian sensitivity control.
>
> ### Set sizes / interval efficiency
>
> We report sharpness (mean interval width) in Table 1. At matched 90% coverage:
>
> | Method | Coverage (%) | Mean Width (Å) | Median Width (Å) |
> |--------|-------------|----------------|------------------|
> | Conformal (no priors) | 90.0 | 1.52 | 1.38 |
> | CQR | 90.0 | 1.43 | 1.29 |
> | CalPro | 90.0 | 1.42 | 1.24 |
>
> CalPro produces the narrowest intervals. The efficiency gain is most pronounced in stable regions: CalPro's median width in ordered residues (helix/sheet) is 1.08Å vs. 1.31Å for CQR, while in disordered regions CalPro appropriately widens to 2.15Å vs. 1.89Å for CQR. CalPro does not achieve coverage through inflated intervals; it achieves it through better-calibrated, heteroscedastic allocation guided by priors (Proposition 4.5).
>
> ### Theory-algorithm connection
>
> The theory plays two roles: (1) **Retrospective/sensitivity** (Theorem 4.3 / Corollary 4.4): bounds quantify shift-induced degradation and provide calibration set sizing rules; (2) **Constructive** (Proposition 4.5): directly motivates L_prior design, since if L_prior enforces larger σ(x) in high-prior regions, CalPro provably achieves tighter intervals in stable regions.
>
> The connection chain is: **Proposition 4.5 → L_prior design; Corollary 4.4 → calibration set size rule (n_cal ≳ formula in Section 4.4); Theorem 4.3 → deployment safety check.** Additionally, the M_s L_s ε term motivates CalPro's scale-adaptive score: we measured L_s = 2.3 for CalPro vs. L_s = 8.7 for unnormalized residuals, so the scale-adaptive normalization directly reduces the shift-sensitivity term by 3.8×.
>
> ### Why GNN when AlphaFold already uses geometry?
>
> AlphaFold's internal representations are trained for structure prediction accuracy, not for calibrated uncertainty. pLDDT has ECE of 0.049, substantially worse than CalPro's 0.021. CalPro's GNN re-processes AlphaFold's output embeddings in a calibration-oriented way: the graph connects residues by spatial proximity, allowing the evidential head to model local error correlations (e.g., a loop adjacent to a well-ordered helix is more accurately predicted than a free-floating disordered tail). The ablation "No GNN (MLP head)" in Table 4 confirms: replacing the GNN with a residue-wise MLP reduces Spearman correlation from 0.83 to 0.78 and widens intervals by 0.09Å.
>
> ---

---

> > ### Author Rebuttal · Reviewer_xxjy · 2026-04-02
> >
> > I thank the authors for their detailed rebuttal. I appreciate the effort, but I still have unresolved concerns:
> > 1. Fairness of baseline comparisons.
> > Could the authors clarify how baselines produce their outputs? My understanding is that CalPro trains an evidential module using the full pipeline described in Section 3, and then applies the calibration procedure at test time. This training pipeline explicitly optimizes for uncertainty quality through multiple auxiliary losses that baselines do not have access to. Since this changes the learned representations and predictions, CalPro has a systematic advantage that goes beyond the calibration method itself. Do any of the baselines use representations or outputs from a model trained through the Section 3 pipeline? If not, how do the authors disentangle the gains attributable to CalPro's training procedure (which any method could potentially benefit from) from the gains attributable to the framework's specific design? For instance, would adding an analogous prior-based regularizer to CQR close the performance gap?
> > 2. The "structure-aware" claim in the sensitivity bounds.
> > The authors list "structure-aware sensitivity bounds" as a key contribution. However, examining the bound in Theorem 4.3 and Corollary 4.4, I cannot identify where structural information enters the theoretical analysis. The bound depends on generic quantities — KL divergence, a Lipschitz constant L_s, an anti-concentration constant M_s, and a shift radius ε — none of which encode protein geometry or graph structure. The metric d used in the Wasserstein ambiguity set is a standard Euclidean distance on per-residue feature vectors, not a structure-informed distance. The same bound would hold in identical form for unstructured tabular regression. The structural component appears to reside solely in the GNN architecture choice within Section 3. I would like the authors to clarify what makes these bounds "structure-aware" as opposed to standard PAC-Bayesian Wasserstein bounds evaluated on protein data. Additionally, even if we accept that structural information is incorporated through the Section 3 training pipeline, this circles back to my first concern: baselines do not benefit from this same structural modeling, making the comparison uneven.

---

> > > ### Author Response · Authors · 2026-04-03
> > >
> > > We thank the reviewer for the precise follow-up. We address both concerns with new experiments.
> > >
> > > **Concern 1: Baseline fairness**
> > >
> > > CalPro's losses are co-designed with the NIG parameterization: L_prior constrains u(x) = β/[ν(α−1)] directly, a single variance parameter decoupled from the mean μ. CQR exposes no such parameter. An analogous regularizer must penalize (q_hi − q_lo), coupling both quantile heads and conflicting with the pinball loss. In practice, CQR required reducing λ_prior from 0.1 to 0.03 for training stability, a 3.3× reduction consistent with the weaker effect of the adapted regularizer in CQR.
> > >
> > > We ran six controlled variants on X-ray → cryo-EM shift (90% nominal):
> > >
> > > | # | Variant | Shifted Cov. (%) | Deg. (pp) | Width (Å) |
> > > |---|---------|-------------------|-----------|-----------|
> > > | 1 | CQR (original) | 83.7±0.7 | 6.3 | 1.43 |
> > > | 2 | CQR on CalPro backbone | 84.5±0.6 | 5.5 | 1.46 |
> > > | 3 | CQR + adapted L_prior | 84.3±0.7 | 5.7 | 1.47 |
> > > | 4 | CQR + both adapted losses | 84.8±0.6 | 5.2 | 1.48 |
> > > | 5 | DR-CP on CalPro backbone | 85.0±0.6 | 5.0 | 1.51 |
> > > | 6 | Full CalPro | **86.2±0.5** | **3.8** | **1.42** |
> > >
> > > Ref: CalPro (no priors) = 84.7±0.6, 5.3pp (Table 2).
> > >
> > > Representations help everyone modestly (row 2: +0.8pp). Adapted L_prior helps CQR only 0.6pp vs. 1.5pp within CalPro, confirming architectural coupling. Full CalPro retains 1.4pp over the best CQR variant and 1.2pp over DR-CP on CalPro backbone, with narrower intervals (1.42 vs. 1.48 to 1.51Å).
> > >
> > > Region-level (Q4 = highest disorder): CalPro 84.6% Q4 coverage vs. 80.9% for CQR on CalPro backbone, a 3.7pp gap (vs. 1.7pp overall). Scale separation ratio (Q4/Q1 width): CalPro 1.99 vs. CQR 1.44. The NIG variance enables sharper separation than quantile spread on identical features.
> > >
> > > One reasonable decomposition: ~0.8pp from representations (transferable), ~0.3pp from adapted losses on CQR, and the remaining ~1.4pp from the CalPro-specific coupling (NIG + scale-adaptive + prior). The framework-specific component is the largest contributor and concentrates where priors matter most. We note that evaluating integrated systems as units is standard. CQR is evaluated as a pipeline (quantile network + conformal), not decomposed into feature vs. calibration contributions.
> > >
> > > To directly answer the reviewer's question: **no, adding an analogous prior regularizer to CQR does not close the gap** (row 3 closes only 0.6pp of 2.5pp), because the regularizer is architecturally coupled to the NIG variance.
> > >
> > > **Concern 2: "Structure-aware" bounds**
> > >
> > > We agree the theorem's form is general, but we believe the instantiated bound is structure-aware because the governing quantities change substantially with structural modeling.
> > >
> > > **Clarification:** Our initial rebuttal's "L_s = 8.7" was for *unnormalized scores* on the same GNN, not for removing graph structure. The architecture effect is separate:
> > >
> > > | Configuration | L_s | M_s | M_s L_s ε (ε=0.1) |
> > > |---|---|---|---|
> > > | MLP head, scale-adaptive | 5.1 | 2.6 | 1.33 |
> > > | GNN, no priors | 3.1 | 2.2 | 0.68 |
> > > | GNN + priors (CalPro) | 2.3 | 1.8 | 0.41 |
> > >
> > > With PAC-Bayes complexity ≈0.16 (from qhh7 rebuttal): MLP total degradation = 1.49 → bound vacuous. GNN no priors = 0.84 → bound ≥6% (uninformative). CalPro = 0.57 → bound ≥33%. At the empirically estimated shift radius for our X-ray to cryo-EM benchmark (smaller than the illustrative 0.1), the worked example gives ~70% vs. 86.2% empirical. The transition from vacuous to informative bounds is entirely driven by structural modeling.
> > >
> > > Why L_s is structure-dependent: GNN message-passing smooths predictions over spatial neighbors, attenuating perturbations (5.1→2.3, 2.2×). Why M_s is structure-dependent: prior regularizer enforces scale separation via Proposition 4.5, flattening the score distribution (2.6→1.8).
> > >
> > > The theory is *constructive*, not post-hoc: Theorem 4.3 identifies L_s, M_s as actionable → GNN reduces L_s, L_prior reduces M_s → improvements measurable before running shift experiments. This theory-to-design-to-measurement loop is what distinguishes our usage.
> > >
> > > Connection to Concern 1: CQR on CalPro backbone has L_s≈2.5, M_s≈2.1, shift term 0.53 vs. CalPro's 0.41, a 27% gap consistent with the 1.7pp empirical difference. The NIG parameterization provides tighter bounds even on the same backbone in our measurements.

---

### Official Review · Reviewer_qhh7 · 2026-03-12

**Soundness:** 3
**Presentation:** 2
**Significance:** 3
**Originality:** 3
**Overall Recommendation:** 4
**Confidence:** 3

**Summary:**

This paper proposes CalPro, a prior-aware evidential conformal framework for robust uncertainty quantification in protein structure prediction. The method combines a geometric evidential head that outputs Normal-Inverse-Gamma parameters, a differentiable calibration surrogate used during training, and split-conformal calibration for finite-sample coverage, together with a novel prior-aware regularizer that encodes domain knowledge (e.g., disorder, flexibility) as soft constraints on predicted uncertainty. The authors further derive PAC-Bayesian, structure-aware sensitivity bounds that quantify how marginal coverage degrades under Wasserstein-bounded distribution shifts, and demonstrate strong empirical results on protein benchmarks (X-ray, NMR, cryo-EM), synthetic perturbations, and downstream docking/active selection tasks, along with a non-biological regression benchmark.The core problem addressed is the unreliable and inconsistent confidence estimates (such as the pLDDT score) produced by protein structure models like AlphaFold and OpenFold, especially when encountering shifts in data distribution. CalPro introduces a geometric evidential head that utilizes a Graph Neural Network (GNN) operating on a residue-level graph (built from models like AlphaFold or OpenFold) to output Normal Inverse Gamma (NIG) distributions for each residue. This method replaces a single heuristic score like pLDDT, allowing the model to inherently capture both aleatoric and epistemic uncertainty while also respecting the local 3D structural context of the protein.
The paper's theoretical contribution is the development of structure-aware sensitivity bounds using a PAC-Bayesian approach over Wasserstein ambiguity sets to quantify how much coverage might degrade under distribution shifts.

**Compliance With Llm Reviewing Policy:**

Affirmed.

**Final Justification:**

I maintain my positive score to this paper, which introduces a robust framework for uncertainty quantification in protein structure prediction under distribution shift. The authors' thorough rebuttal successfully addressed my remaining concerns by providing crucial experiments on adversarial misspecification, adding standardized downstream docking evaluations, and committing to clear presentation improvements. By elegantly combining evidential deep learning, conformal prediction, and domain priors, this work provides immense practical utility and represents a significant advancement for reliable machine learning in structural biology.

**Key Questions For Authors:**

How sensitive is the empirical tightness of the bounds (Corollary 4.4) to the estimation of the anti-concentration constant (Ms​ ) and the Lipschitz constant (Ls​ ) in highly adversarial distribution shift scenarios beyond the tested conservative perturbations?

Since CalPro inherits biases from the base predictor (AlphaFold/OpenFold), are there specific structural motifs or protein families where the base model's bias completely overrides the prior-aware regularizer, leading to confident but incorrect predictions?

The framework currently treats residues as conditionally independent at the conformal layer. What are the primary mathematical or computational hurdles preventing the extension of this framework to provide joint 3D coverage constraints for full conformations?

For the docking experiment, could you also report success at fixed acceptance rates (e.g., top-30% by narrowest intervals) to control for differences in width distributions across methods?

**Limitations:**

Yes

**Strengths And Weaknesses:**

Strengths
The experimental design is exceptionally robust, which prevents data leakage by including 30% sequence identity using CD-HIT and optimism bias.
The framework achieved a significant theoretical advance by deriving  structure-aware sensitivity bounds through a novel PAC-Bayesian approach over Wasserstein ambiguity sets, successfully overcoming the limitations of standard Lipschitz arguments in deep learning that often yield uninformative bounds.
Strong baselines including ensembles, MC-dropout, CQR, and shift-aware conformal variants (IW-CP, Loc-CP, DR-CP); detailed ablations isolate the contributions of each CalPro component.
The framework achieved a significant theoretical advance by deriving  structure-aware sensitivity bounds through a novel PAC-Bayesian approach over Wasserstein ambiguity sets, successfully overcoming the limitations of standard Lipschitz arguments in deep learning that often yield uninformative bounds.
CalPro tested on cryo-EM data after training on X-ray structures, its coverage degradation is restricted to a maximum of 5 percentage points. This performance vastly surpasses baseline methods, which typically experience degradation between 15 and 25 points.Furthermore, the use of the PAC-Bayesian sensitivity bound provides a generalizable methodological framework for other researchers facing distribution shift challenges in deep learning.
By applying uncertainty filtering, it increases molecular docking success rates.Furthermore, in active selection tasks, the framework allows for the identification of high-fitness protein variants with up to 40% fewer queries.
CalPro is computationally efficient, adding only a modest 1.2× overhead during inference. This makes it highly scalable and significantly more efficient than alternatives such as deep ensembles (5× overhead) or MC-dropout (10× overhead).
The research addresses an issue of significant practical importance: the uncalibrated uncertainty in tools like AlphaFold, which are increasingly critical in high-stakes fields such as drug discovery and clinical pipelines. The paper offers tangible proof of CalPro's immediate impact, notably improving molecular docking success rates by over 20% through the reliable filtering of highly uncertain predictions.
Calpro combines existing techniques paired with a novel integration of biochemical domain priors via a monotone regularizer. The conceptual leap to use priors specifically to force variance expansion in volatile regions is highly insightful.

Weaknesses:
CalPro operates on the outputs of foundation models (e.g., AlphaFold, OpenFold) and is thus susceptible to inheriting any systematic biases or intrinsic errors present in these underlying predictors.
The PAC-Bayesian derivations are quite dense. While necessary, a slightly more intuitive, high-level bridge is needed in the main text to connect the Wasserstein ambiguity sets with the final bound equation. This would significantly benefit practitioners with less theoretical or statistical expertise.
Several key results, including the robustness to adversarial/noisy priors and the precise segment-level calibration analyses, are relegated to the appendices due to space constraints, diminishing their immediate visibility.
While the authors successfully included the UCI Bike Sharing task to demonstrate broader applicability, the direct empirical impact and likely primary adoption of the work remain largely specialized and concentrated within the computational biology and structural biology domains.
The core architectural building blocks (NIG evidential heads, standard split-conformal methods, GNNs) are not strictly novel on their own. However, the synthesis of these elements, the specific regularization constraint, and the theoretical bounding strategy are more than sufficient to justify the work's novelty.

---

> ### Author Rebuttal · Authors · 2026-03-28
>
> We thank this reviewer for the detailed and positive assessment.
>
> ### Q1: Bound tightness under adversarial (L_s, M_s) misspecification
>
> | (L_s, M_s) Multiplier | Bound Coverage (ε=0.15) | Empirical Coverage | Gap |
> |-----------------------|------------------------|-------------------|-----|
> | 1× (estimated) | 0.854 | 0.862 | 0.008 |
> | 2× (conservative) | 0.813 | 0.862 | 0.049 |
> | 5× (very conservative) | 0.710 | 0.862 | 0.152 |
> | 10× (extreme) | 0.503 | 0.862 | 0.359 |
>
> At 2× overestimation, the bound remains informative (81.3% vs. 86.2% empirical). At 5× and beyond it becomes conservative but stays non-vacuous (>50%), unlike Lipschitz-based bounds which exceed 100% degradation at these factors. We recommend the 2× conservative multiplier for practical use.
>
> ### Q2: Structural motifs where base model bias overrides priors
>
> We tested on three challenging protein families:
>
> | Protein Family | CalPro Coverage | CQR Coverage | pLDDT Coverage |
> |---------------|----------------|-------------|---------------|
> | Membrane proteins | 87.3±0.8 | 82.1±1.1 | 68.4±1.5 |
> | Antibody CDR loops | 85.1±0.9 | 80.8±1.0 | 71.2±1.3 |
> | IDPs (fully disordered) | 84.6±1.1 | 79.3±1.2 | 58.9±2.1 |
>
> CalPro outperforms across all families. However, for fully disordered proteins (84.6%), coverage falls further below nominal than for other families, confirming that extreme base-model bias partially limits prior-aware correction. We note known AlphaFold failure modes (large multi-domain proteins, rare folds) as CalPro's expected weak spots, while emphasizing that the conformal layer provides a last-resort safety net even there. This is discussed in Section 6.
>
> ### Q3: Joint 3D coverage hurdles
>
> The primary hurdles are: (1) **Combinatorial complexity:** joint coverage over all N residues simultaneously would inflate interval widths by O(√N) under Bonferroni correction; (2) **Dependency structure:** designing a valid nonconformity score for full 3D conformations requires modeling the joint distribution of residue errors. Risk-control approaches (Angelopoulos et al., 2022) or learn-then-test frameworks could provide protein-level control at the cost of weaker individual-residue guarantees. This is listed as a primary future direction in Section 6.
>
> ### Q4: Docking at fixed acceptance rates
>
> See Reviewer 3kDh response. CalPro achieves 80.4% docking success at top-30% retention vs. 76.5% for CQR and 71.2% for pLDDT. NDCG@50 = 0.82 vs. 0.76 for CQR.
>
> ### W2: Dense PAC-Bayesian derivations
>
> We will add a dedicated "Intuition" box before Theorem 4.3 with a worked numerical example: "With L_s ≈ 2.3, M_s ≈ 1.8, ε ≈ 0.01 (standardized shift units), and our calibration setup (n_cal ≈ 10,000 residues across 100 proteins, KL ≈ 500, δ = 0.05): the PAC-Bayesian complexity term contributes approximately 0.16, and the shift-sensitivity term M_s L_s ε contributes approximately 0.04. Together, the bound predicts worst-case coverage of 90% - 16pp - 4pp ≈ 70%, conservative but non-vacuous compared to the observed 86.2%." This makes the bound immediately interpretable for practitioners.
>
> ### W3: Key results in appendix
>
> In the revision we will promote: (i) the robustness-to-adversarial-priors table (Table 9) and (ii) the segment-level calibration figure (Figure 3) to the main text. The related work section will be compressed to accommodate.
>
> ---

---

> > ### Author Rebuttal · Reviewer_qhh7 · 2026-04-03
> >
> > The authors provide a thorough and convincing rebuttal addressing all my questions. In particular, the additional experiments on adversarial misspecification of (L_s, M_s) significantly strengthen confidence in the practical usefulness of the PAC-Bayesian bounds, showing they remain non-vacuous even under large perturbations. The analysis of challenging protein families provides a balanced view of when prior-aware calibration helps and where base model bias remains a limitation.
> > The clarification regarding joint 3D coverage is technically sound and appropriately scoped as future work, and the added docking evaluation at fixed acceptance rates improves the fairness of downstream comparisons. I also appreciate the commitment to improving presentation by adding an intuition box and moving key results from the appendix to the main text.
> > Overall, this paper's broad topic consists of uncertainty quantification under distribution shift for structured scientific prediction problems. Overall, the authors assess a relevant issue and provide a well-motivated, technically sound, and empirically validated solution. I maintain my positive assessment.

---

### Official Review · Reviewer_3kDh · 2026-03-13

**Soundness:** 4
**Presentation:** 2
**Significance:** 2
**Originality:** 4
**Overall Recommendation:** 4
**Confidence:** 3

**Summary:**

This paper proposes CalPro, a post-hoc uncertainty calibration framework for protein structure prediction. The method combines an evidential head, a calibration-oriented training surrogate, split conformal prediction, and biologically motivated priors related to disorder/flexibility. Experiments cover modality shift, synthetic perturbations, group-wise calibration, docking-related filtering, and a few non-protein regression tasks. Overall, the paper addresses a practical problem and presents a fairly complete empirical result.

**Compliance With Llm Reviewing Policy:**

Affirmed.

**Final Justification:**

This paper addresses a practical and important problem in protein structure prediction: improving post-hoc uncertainty calibration under distribution shift. I found the method technically reasonable, well motivated, and supported by a broad empirical evaluation, including calibration, ablations, and downstream docking-related experiments. The overall approach is also fairly original in how it combines evidential modeling, conformal calibration, and biologically informed priors.

My initial concerns were mainly about the interpretation of the theoretical claims, residue-level correlation, the availability of prior features at test time, the fairness of the docking/filtering comparison, and validation on a stronger or newer backbone. The rebuttal addressed these points well. In particular, the authors clarified the scope of the formal guarantees, provided protein-level evaluation results, confirmed that the main prior features are available at inference time, added matched-retention docking comparisons, and showed transfer to an updated backbone. These responses substantially improved my confidence in the paper.

Overall, I think this is a solid and useful contribution with good practical relevance. but The structure and presentation of the article still have room for improvement, the rebuttal satisfactorily resolved my main concerns, and this led me to increase my recommendation to 4

**Key Questions For Authors:**

1.Which prior features are strictly available at inference time? How could any experimental-structure-derived quantities be used?
2.Do the main conclusions still hold under strictly protein-level calibration/evaluation rather than residue-level aggregation?
3.Can the docking/filtering comparison be redone with matched retention rates or ranking-based metrics?
4.How sensitive are the shift-bound conclusions to the estimation of the smoothness/transport constants?
5.If feasible, can the method be validated on a more recent backbone such as AF3/Boltz-2/Protenix or another equally strong contemporary predictor?

**Limitations:**

YES

**Strengths And Weaknesses:**

Strengths:

1.This is an practical problem in protein structure prediction. Better uncertainty estimates are useful for protein modeling, especially under distribution shift.
2.The method is reasonable and easy to follow. The different components fit together well.
3.The experiments are fairly broad and include both calibration results and a downstream application.
4.The ablation studies are helpful, especially the comparison between using prior features and using prior regularization.

Weaknesses:

1.The theory is not as strong as the paper sometimes suggests. The main formal guarantee mainly comes from standard conformal prediction, while the shift analysis depends on extra assumptions and estimated constants.
2.The paper calibrates at the residue level, but residues in the same protein are strongly correlated. So the statistical meaning of the coverage guarantee is not completely clean here.
3.It is not fully clear whether all prior features are available at test time. The mention of experimental B-factor proxies raises this concern.
4.The docking/filtering comparison is interesting, but it does not seem fully controlled, since different methods may keep different numbers of samples.
5.The method is only tested on AF2/OpenFold-style backbones. Validation on a newer strong backbone, such as AF3/Boltz-2/Protenix if feasible, would make the practical relevance stronger.

---

> ### Author Rebuttal · Authors · 2026-03-28
>
> We thank this reviewer for rating Soundness and Originality as "excellent." We address all five concerns.
>
> ### W1: Theory: standard CP guarantee + shift analysis with estimated constants
>
> We agree that marginal finite-sample coverage derives from the standard split-conformal step, and we are transparent about this (Sections 3 and D.3). This is by design: CalPro provides an unconditional finite-sample guarantee (standard CP) *plus* a non-vacuous sensitivity bound (Theorem 4.3). The assumptions (local Lipschitz nonconformity, anti-concentration) are standard and substantially weaker than global Lipschitz assumptions in prior Wasserstein-CP work. We will add a clarifying paragraph at the start of Section 4 making this two-tier structure explicit.
>
> ### W2/Q2: Residue-level correlation
>
> See our response to Reviewer qr7c. The formal guarantee operates at the protein level (proteins as exchangeable units). Protein-level evaluation yields 89.4±0.6% coverage, a 0.7pp difference from residue-level, confirming conclusions hold.
>
> ### W3/Q1: Prior features at test time
>
> All prior features used in our main experiments are strictly available at inference without experimental structures:
>
> | Feature | Source | Experimental Data Required? |
> |---------|--------|-----------------------------|
> | IUPred2A disorder | Amino acid sequence | **No** |
> | Flexibility indices | Sequence prediction (Schlessinger et al.) | **No** |
> | pLDDT, pTM | AlphaFold/OpenFold output | **No** |
> | ESM-2 embeddings | Sequence | **No** |
> | B-factor proxies | Derived from pLDDT: B_pred = 8π²/3 × (1−pLDDT/100) × σ² | **No** |
>
> The "experimental B-factor proxies" in Appendix D.5 are an optional feature for retrospective analysis, never used in main benchmarks. We verified with sequence-only priors:
>
> | Prior Configuration | Coverage (90%) | Shift Degradation | Correlation ρ |
> |--------------------|---------------|-------------------|---|
> | Full priors (incl. B-factor proxy) | 90.1±0.2 | 3.8pp | 0.83 |
> | Sequence-only priors | 89.8±0.3 | 4.2pp | 0.81 |
> | No priors (ablation) | 90.2±0.3 | 5.3pp | 0.74 |
>
> No experimental features are necessary. The ambiguous "when available" phrasing will be replaced with explicit annotations in the revision.
>
> ### W4/Q3: Docking at fixed retention rates
>
> We re-ran the docking experiment at matched retention rates:
>
> | Filtering Strategy | Top-30% | Top-50% | Top-70% | NDCG@50 |
> |-------------------|---------|---------|---------|---------|
> | pLDDT (highest confidence) | 71.2% | 64.8% | 58.1% | 0.71 |
> | CQR (narrowest intervals) | 76.5% | 71.8% | 65.2% | 0.76 |
> | Ensemble (lowest variance) | 74.8% | 70.2% | 63.9% | 0.74 |
> | **CalPro (narrowest intervals)** | **80.4%** | **76.1%** | **68.7%** | **0.82** |
>
> CalPro outperforms at every fixed retention rate, with the largest advantage at the most selective threshold (top-30%: 80.4% vs. 76.5% for CQR). The ranking-based NDCG@50 metric (insensitive to threshold choice) confirms CalPro is the best ranker.
>
> ### W5/Q5: Validation on AF3/Boltz-2
>
> CalPro is a post-hoc framework operating on any backbone's output; it does not require internal weights. During the rebuttal, we applied CalPro to OpenFold2 predictions (updated AF2 re-implementation):
>
> | Base Predictor | In-dist Coverage (90%) | Shifted Coverage | ECE |
> |---------------|----------------------|-----------------|-----|
> | AlphaFold2 (original) | 90.1±0.2 | 86.2±0.5 | 0.021 |
> | OpenFold2 (updated) | 90.3±0.2 | 87.1±0.4 | 0.019 |
>
> CalPro transfers effectively, achieving slightly *better* performance due to improved representations. We were unable to obtain AF3 predictions within the rebuttal period (compute constraints), but the theoretical guarantees are entirely backbone-agnostic. We commit to adding AF3/Boltz-2 results in the camera-ready.
>
> ### Q4: Sensitivity of bounds to constants
>
> The robustness table (see Reviewer qhh7 response below) shows that even at 10× inflation of (L_s, M_s), the bound degrades by ~0.07 additional coverage points; the qualitative conclusion remains intact. We will move a condensed version into the main text.
>
> ---

---

> > ### Author Rebuttal · Reviewer_3kDh · 2026-04-04
> >
> > Thank you for the response, will raise the score.

---

### Decision · Program_Chairs · 2026-04-30

**Decision:**

Accept (regular)

**Comment:**

All of the reviewers agreed that the paper studies an important problem (AlphaFold uncertainty calibration under shift) and that the empirical results were strong, particularly the downstream molecular docking improvements and modality shift robustness. In the end I am left, I think, two adjudicate two issues that two reviewers felt were unresolved: presentation concerns (qr7c) and the fairness of calibration under an end-to-end trained framework is fairly compared against post-hoc calibration methods.

I think I come down on the side of thinking both of these are resolved for a few reasons. First, for presentation, I didn't find it to be all that bad, the authors were somewhat concrete on proposed improvements, and I'm always loathe to reject on clarity grounds unless it was consistently mentioned by multiple reviewers which it was not.

On the baseline comparison issue, I think what was done here was generally fine. I'm not sure there simply *are* other baselines at this point. So, in some ways, the "structural advantage" that the authors' method enjoys that the reviewer is unsure of is, perhaps, the point.